

# 2-group global symmetries, hydrodynamics and holography

Nabil Iqbal[1*] and Napat Poovuttikul[2†]

**1** Centre for Particle Theory, Department of Mathematical Sciences, Durham University,
South Road, Durham DH1 3LE, UK
**2** Science Institute, University of Iceland, Dunhaga 3, IS-107, Reykjavik, Iceland

* nabil.iqbal@durham.ac.uk , † nickpoovuttikul@hi.is

## Abstract

2-group global symmetries are a particular example of how higher-form and conventional global symmetries can fuse together into a larger structure. We construct a theory of hydrodynamics describing the finite-temperature realization of a 2-group global symmetry composed out of $U(1)$ zero-form and $U(1)$ one-form symmetries. We study aspects of the thermodynamics from a Euclidean partition function and derive constitutive relations for ideal hydrodynamics from various points of view. Novel features of the resulting theory include an analogue of the chiral magnetic effect and a chiral sound mode propagating along magnetic field lines. We also discuss a minimalist holographic description of a theory dual to 2-group global symmetry and verify predictions from hydrodynamic descriptions. Along the way we clarify some aspects of symmetry breaking in higher-form theories at finite temperature.

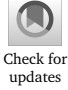

## 1  Introduction

It is a remarkable fact that the long-distance physics of almost any interacting quantum system at finite temperature is given by the theory of hydrodynamics. The ubiquity of such a description essentially arises from the existence of conserved quantities: the conserved density of a charge (such as a number current, or a momentum) cannot change arbitrarily fast, as any local change in the density must be supplied by a flow of current. The resulting slow dynamics of the charges is governed by the universal framework of hydrodynamics [1, 2].

As the existence of hydrodynamics is intimately intertwined with the structure of conserved quantities, *global* symmetries always play a crucial role in formulating a hydrodynamic description. Recently, the set of systems admitting such a universal hydrodynamic description have been enlarged, due to the emphasis and systematic studies of, *higher form global symmetries* [3]. Just as an ordinary $U(1)$ global symmetry enforces the existence of a conserved particle number with a current $j^\mu$, a higher-form global symmetry enforces the existence of a conserved density of higher-dimensional objects, such as strings (or branes). More specifically, a $p$-form symmetry results in a conserved current that is a $p + 1$-form: thus a conventional current $j^\mu$ is associated with a 0-form symmetry. One extremely familiar example of a 1-form symmetry is Maxwell electrodynamics in four dimensions – the symmetry controlling the conservation of magnetic flux is precisely such a higher-form symmetry, with associated conserved current $J^{\mu\nu} = \frac{1}{2}\epsilon^{\mu\nu\rho\sigma}F_{\rho\sigma}$. The realization of this higher-form symmetry at finite temperature can be shown to lead to a hydrodynamic theory which is a reformulation of dissipative relativistic magnetohydrodynamics [4–6].[1]

A new set of interesting phenomena are possible when the theory has both an ordinary 0-form global symmetry with conserved current $j^\mu$ and a 1-form global symmetry with conserved current $J^{\mu\nu}$. It is now possible for the resulting symmetries to *intertwine* in a non-trivial manner characterized by an integer coefficient $\hat{\kappa}$. There are many ways to describe the resulting structure, called a *2-group*: one simple implication is that the ordinary $U(1)$ current is not conserved in the presence of *both* an external source and a dynamical 1-form charge. The Ward identity is:

$$\nabla_\mu \langle j^\mu \rangle = \frac{\hat{\kappa}}{2}\langle J^{\mu\nu}\rangle F_{\mu\nu}, \qquad \nabla_\mu \langle J^{\mu\nu}\rangle = 0. \tag{1}$$

Here $F = da$, where $a_\mu$ is an external gauge field source that couples to the current operator $j^\mu$. The current algebra is modified, but the symmetries formally remain non-anomalous, by which we mean that the partition function as a function of the sources remains invariant under

---

[1]At ideal hydrodynamic level, the dynamic of 2-form current has been discussed in e.g. relativistic magneto-hydrodynamic literature [7] and effective dynamics of higher-dimensional black holes [8]. At dissipative level, it was discussed in [9] but without connection to higher-form symmetry.

an appropriate symmetry transformation.[2] One example where this structure occurs is $U(1)$ gauge theories coupled to appropriate fermionic matter. Consider, for example, a theory with ordinary $U(1)_1 \times U(1)_2$ 0-form global symmetry with a mixed anomaly of the usual kind in the $U(1)_1$-$U(1)_1$-$U(1)_2$ sector. The symmetry structure (1) can be obtained by *gauging* the $U(1)_2$ non-anomalous subgroup of the global symmetry in such theory, namely by couple it to a dynamical gauge field of the $U(1)_2$. The first Ward identity is nothing but the anomalous Ward identity of the ungauged $U(1)_1$ current $j^\mu$ with the gauged magnetic flux $(F_2)_{\mu\nu}$ replaced by the dynamical magnetic flux operator $J = \star F_2$ namely

$$\partial_\mu \langle j^\mu \rangle = -\frac{1}{2}\kappa_{a^2 v}\epsilon^{\mu\nu\rho\sigma}(F_1)_{\mu\nu}(F_2)_{\rho\sigma} \qquad \underset{\text{gauging } U(1)_v}{\Longrightarrow} \qquad \partial_\mu j^\mu = \frac{1}{2}\hat{\kappa}\langle J^{\mu\nu}\rangle(F_1)_{\mu\nu}.$$

Here, the 2-group structure constant $\hat{\kappa}$ is determined by a coefficient of the mixed anomaly between $U(1)_1$ and $U(1)_2$. On the other hand, the conservation of $\langle J^{\mu\nu}\rangle = \langle\star(F_2)^{\mu\nu}\rangle$ is nothing but conservation of magnetic flux constructed from $U(1)_2$ gauge field, and remains exact.

Gauging a subgroup of an anomalous theory is just one of many ways to obtain the 2-group structure. In fact, 2-group or higher-group is a special kind of category which enlarges the concept of group to a higher dimensional algebra [10]; we found Ref. [11] to be a good entry point into the rather extensive literature. Unlike the mixed anomaly example above that exists only in $3 + 1$ dimensions, a 2-group global symmetry can occur in any dimension larger than $1 + 1$ such that $J^{\mu\nu}$ has a continuous value. There are various interesting quantum field theories where the 2-group structure have been discussed see e.g. [12–23] and references therein.[3]

Here we emphasize that 2-group is a genuine global symmetry structure of the theory and should be treated as such. In particular, the 2-group symmetry can be encoded in the generating function

$$Z[g_{\mu\nu}, a_\mu, b_{\mu\nu}] = \left\langle \exp\left[i \int d^{d+1}x \sqrt{-g}\left(\frac{1}{2}T^{\mu\nu}g_{\mu\nu} + j^\mu a_\mu + \frac{1}{2}J^{\mu\nu}b_{\mu\nu}\right)\right]\right\rangle, \qquad (2)$$

where $a_\mu, b_{\mu\nu}$ are the source for an ordinary conserved current $j^\mu$ the higher-form current $J^{\mu\nu}$ respectively. The 2-group Ward identities (1) can be obtained if the generating function is invariant under the following transformation

0-form $U(1)$ transformation : $\qquad a_\mu \to a_\mu + \partial_\mu\lambda, \qquad b_{\mu\nu} \to b_{\mu\nu} + \hat{\kappa}(da)_{\mu\nu}\lambda, \qquad$ (3a)

1-form $U(1)$ transformation : $\qquad b_{\mu\nu} \to b_{\mu\nu} + 2\partial_{[\mu}\Lambda_{\nu]}, \qquad\qquad$ (3b)

which can be thought of as a Green-Schwarz mechanism for the background fields [14, 19, 21].[4] A more precise statement for discrete groups can be found in e.g. [21].

In this work, we will focus on a class of gapless theories in the IR which realized 2-group global symmetry at *finite temperature and densities*, i.e. in a highly dynamical regime where we expect hydrodynamics to be applicable. We have two main motivations. Firstly, it is intrinsically interesting to systematically expand the hydrodynamic framework to describe new classes of interesting quantum field theories with more intricate symmetry structures. This is

---

[2]Here, to distinguish clearly between operators and sources we have denoted the operators by their expectation values, but the relation above holds in all states and is an operator equation.

[3]The readers can find a nice introduction to 2-group from the current algebra and charge multiplication approach in [19] and [21] respectively.

[4]To be very precise, the transformation in (3) does not capture all elements of 2-group. For a 2-group constructed from a zero-form $G$ and an abelian one-form $\mathfrak{a}$ global symmetry, it can be defined by a cross module $(G, \mathfrak{a}, \rho, \hat{\kappa})$. Here $\hat{\kappa}$ is an integer appearing in (3) and $\alpha$ is a map from $G$ to a group automorphism of $\mathfrak{a}$, which is trivial in our setup. This version is called coherent 2-group [10] which is a specification of a more general definition found in Ref. [11]. More physical implications and examples of $\alpha$ and $\hat{\kappa}$ can be found in e.g. [14, 21].

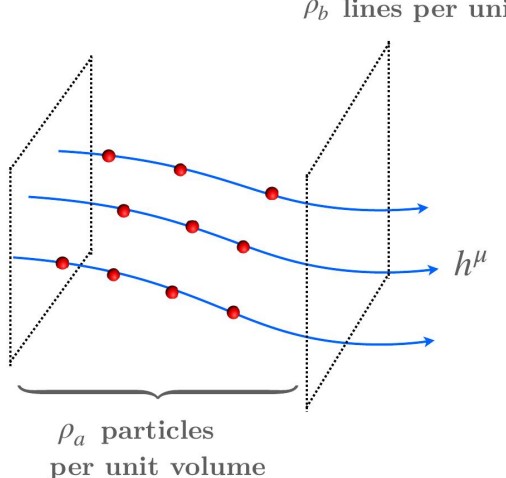

Figure 1: Illustration of $\rho_b$ strings per unit area pointing along the direction of a spatial unit vector $h^\mu$ combined with $\rho_a$ particle densities. This ensemble arrange itself in such a way that the 2-group Ward identities are satisfied.

in the same spirit as many of the earlier mentioned works, particularly [14, 17, 20, 22], which apply the concept of 2-group to enlarge the description of IR gapped phases with non-trivial topological properties. Secondly, it could also serve as a toy model to better understand the magnetohydrodynamic limit of chiral medium, obtained by gauging the vector $U(1)$ subgroup of a system with an ABJ anomaly [24–27]. Though this problem turns out to have a different character, we will comment further on this aspect in the discussion.

Below, we present a framework to consistently construct the hydrodynamic description of systems with 2-group global symmetry. We give a summary of our results and some technical motivations as well as an outline of the remainder of the manuscript in the next section.

## 1.1  Summary and outline

We argue that in the ideal limit the constitutive relations for a theory with 2-group global symmetry with *unbroken* zero-form $U(1)$ symmetry, namely $\langle T^{\mu\nu}\rangle, \langle J^{\mu\nu}\rangle, \langle j^\mu\rangle$ expressed in terms of fluid variables, are

$$\langle T^{\mu\nu}\rangle = (\varepsilon + p)u^\mu u^\nu + p g^{\mu\nu} - \mu_b \rho_b h^\mu h^\nu - \hat{\kappa}\mu_a^2 \rho_b(u^\mu h^\nu + u^\nu h^\mu), \tag{4a}$$

$$\langle J^{\mu\nu}\rangle = \rho_b(u^\mu h^\nu - u^\nu h^\mu), \tag{4b}$$

$$\langle j^\mu\rangle = \rho_a u^\mu - 2\hat{\kappa}\mu_a \rho_b h^\mu - \hat{\kappa}\langle J^{\mu\nu}\rangle a_\nu, \tag{4c}$$

where $u^\mu$ plays a role of fluid velocity and $h^\mu$ is a unit vector pointing the direction of the string-like charged objects. Both vectors are normalised, $u^\mu u_\mu = -1, h^\mu h_\mu = 1$, and are orthogonal to one another $u^\mu h_\mu = 0$. The chemical potentials $\mu_a, \mu_b$ and the densities $\rho_a, \rho_b$ with subscript $a, b$ are associated to the zero-form and one-form $U(1)$ symmetry respectively. Together with the pressure $p$ and the energy density $\varepsilon$, they satisfy the following thermodynamic relations

$$\varepsilon + p = sT + \mu_a \rho_a + \mu_b \rho_b, \qquad dp = s dT + \rho_a d\mu_a + \rho_b d\mu_b, \tag{5}$$

where $p = p(T, \mu_a, \mu_b, \hat{\kappa})$ is an equation of state of a generic thermal equilibrium with 2-group global symmetry. Classically, it is a collections of 'point particle' and 'strings', respectively charged under the zero-form and one-form $U(1)$, as illustrated in Figure 1.

One might be alarmed by the explicit dependence of the background gauge field $a_\mu$ in the constitutive relation (4c) despite the partition function (2) being invariant. This is due

to the non-trivial transformation of $b_{\mu\nu}$ under the zero-form $U(1)$ which depends on the field strength of $a_\mu$ (see (3a)). In fact, one can see that the redefinition of the background gauge field after variation w.r.t. $a_\mu$ yield

$$\langle j^\mu[a]\rangle = \frac{\delta}{\delta a_\mu}\log Z\bigg|_{a\neq 0} \rightarrow \langle j^\mu[a+d\lambda]\rangle = \langle j^\mu[a]\rangle - \hat{\kappa}\langle J^{\mu\nu}\rangle\partial_\nu\lambda. \tag{6}$$

This situation is analogous to the consistent current in anomalous theory [28] (see also [29] for a recent review). It is therefore convenient to define an analogue of a covariant current as

$$j^\mu_{cov} = \langle j^\mu\rangle + \hat{\kappa}\langle J^{\mu\nu}\rangle a_\nu. \tag{7}$$

Together $\langle T^{\mu\nu}\rangle, \langle J^{\mu\nu}\rangle, j^\mu_{cov}$ satisfy the following Ward identity

$$\nabla_\mu\langle T^{\mu\nu}\rangle = \frac{1}{2}H^\nu{}_{\rho\sigma}\langle J^{\rho\sigma}\rangle + F^\nu{}_\mu j^\mu_{cov}, \tag{8a}$$

$$\nabla_\mu\langle J^{\mu\nu}\rangle = 0, \tag{8b}$$

$$\nabla_\mu j^\mu_{cov} = \hat{\kappa}\langle J^{\mu\nu}\rangle F_{\mu\nu}. \tag{8c}$$

The Ward identities (8a)-(8c) and the constitutive relation (4a)-(4c) can be used to extract observable effect such as conserved currents correlation functions at finite temperature and spectrum of collective excitations. Interestingly, the constitutive relation (4a) and (4c) indicate that, even when $a_\mu = 0$ and the system in equilibrium configuration (with $u^\mu = (1, \mathbf{0})$ be the timelike Killing vector), it acquire finite values of $\langle T^{\mu\nu}\rangle$ and $\langle j^\mu\rangle$ at finite chemical potentials due to the 2-group structure constant $\hat{\kappa}$, namely

$$\langle j^\mu\rangle h_\mu\bigg|_{a\to 0} = -2\hat{\kappa}\mu_a\rho_b, \qquad \langle T^{t\nu}\rangle h_\nu = -\hat{\kappa}\mu_a^2\rho_b, \tag{9}$$

where $h_\mu$ is the vector pointed along the direction of the strings (that are counted by $\int \star J$). The equilibrium current is very similar to that arising in anomaly induced transport [30], such as chiral magnetic/vortical effect discussed in [31–33] (see also [34] for a recent review and source of references for this extensive literature). The existence of a dissipationless charged current in equilibrium is somewhat interesting; such phenomena are familiar from e.g. the quantum hall effect, where one finds a similar current flowing around the edge of the sample at a finite chemical potential (see e.g. [35] for a recent discussion). We stress that the microscopic origin of our current is the same as in the regular chiral magnetic effect: there are charged particles flowing along the magnetic field lines. However an important difference in the effective theory is that the magnetic field is not a fixed external source but rather directly the conserved charge $J^{0i}$ that is associated dynamical 2-form current. Thus the current must appear at the level of ideal hydrodynamics, and not at first order as in the conventional chiral magnetic effect.[5]

We find two particularly interesting collective modes around the equilibrium configuration with $h^\mu = \delta^{\mu z}$. The first one, $\omega_\perp(q_z)$ governed the transverse fluctuations along the direction of the string

$$\omega_\perp = \left(-\frac{\hat{\kappa}\mu_a^2\rho_b}{\varepsilon+p} \pm \sqrt{\mathcal{V}_A^2 + \left(\frac{\hat{\kappa}\mu_a^2\rho_b}{\varepsilon+p}\right)^2}\right)q_z, \qquad \mathcal{V}_A^2 = \frac{\mu_b\rho_b}{\varepsilon+p}, \tag{10}$$

---

[5]There are other examples in hydrodynamics where the current $j^\mu$ is misaligned with the fluid velocity $u^\mu$ even at ideal order in derivatives: e.g. a $(1+1)$-d fluid with anomalous $U(1)$ symmetry [35,36], or a $(2+1)$-d example with mixed anomaly between zero-form $U(1)$ and 1-form $U(1)$ [37].

where $q_z$ is the momentum along the direction of $h^\mu$. In the case of $\hat{\kappa} = 0$ and where $J^{ti}$ is the magnetic flux density, this mode $\omega_\perp$ is the Alfvén wave, $\omega_\perp = \pm \mathcal{V}_A q_z$, which is a transverse fluctuations along the magnetic field line. With a finite $\hat{\kappa}$ and $\mu_a$, we can see that the speed of the 'Alfvén wave' is skewed depending on whether the wave propagate in the same or opposite direction of the magnetic flux line i.e. whether $J^{ti}k_i = \pm 1$. The second mode is more complicated but in an equation of state where $\rho_a, \mu_a$ vanish in equilibrium but with finite susceptibility $\chi_{aa} = \partial \rho_a / \partial \mu_a$, we find the following mode governs the longitudinal zero-form $U(1)$ current $\langle j^\mu \rangle h_\mu$

$$\omega_\parallel = -\frac{\hat{\kappa} \rho_b}{\chi_{aa}} q_z \,. \tag{11}$$

This mode is absent in the theory where the global symmetry is a simple product between a zero-form $U(1)$ and a one-form $U(1)$.

We also build a minimal holographic model dual to a QFT with 2-group global symmetry. The bulk fields are gauge fields $\mathcal{A}_a, \mathcal{B}_{ab}$ that encode the data of $(a_\mu, \langle j^\mu \rangle)$ and $(b_{\mu\nu}, \langle J^{\mu\nu} \rangle)$ governed by an action that is invariant under the bulk 2-group gauge transformation:

$$S_{bulk} = -\int d^{d+2}X \sqrt{-G} \left( \frac{1}{4} \mathcal{F}_{ab} \mathcal{F}_{ab} + \frac{1}{6} \mathcal{H}_{abc} \mathcal{H}^{abc} \right), \tag{12}$$

where $\mathcal{H} = d\mathcal{B} - \hat{\kappa} \mathcal{A} \wedge d\mathcal{A}$ is the 2-group field strength. We propose a holographic dictionary as well as associate the macroscopic degrees of freedom with Wilson lines and surfaces in the bulk, explaining their transformation properties. We check that this model exhibits the equilibrium current $j^\mu_{cov} = 0$ for general density $\rho_a, \rho_b$. The spectrum of the chiral mode at $\rho_a = 0$ discussed above is found in the quasinormal mode of the holographic model, and we confirm that its speed of sound agrees with the hydrodynamic prediction.

Readers who are familiar with the hydrodynamic description of a theory with an 't Hooft anomaly may notice more similarities between those and the above construction, despite the fact that the latter is anomaly-free. We shall point out a few notable differences below:

- 't Hooft anomaly induced transport in $2n$ spacetime dimensions occur at $n-1$ order in the gradient expansion, see e.g. [36]. 2-group can only exist in spacetime dimension larger than $d + 1 = 2$ and the new term in the constitutive relation it induced always appear at zeroth order in the derivative expansion.

- In $d + 1 = 4$, one can obtain constitutive relations similar to 2-group by 'weakly gauged' one of the $U(1)$ in the theory with mixed $U(1) \times U(1)$ anomalous global symmetry. This can be done at the cost of the gradient expansion i.e. by promoting the background field strength, originally treated as the first derivative quantity, to the expectation value, which is a zeroth derivative quantity.[6] The procedure presented here requires no such complication; it uses only general principles of effective field theory, and is applicable when $d+1$ is odd and the 'ungauged' anomalous hydrodynamical picture is not available.

The remainder of the paper is dedicated to derivation of the results mentioned above. In section 2, we discuss the thermodynamics of such theories by studying the realization of 2-group global symmetry on the Eucldiean finite-temperature manifold $S^1 \times \mathbb{R}^3$. In section 3, we construct the effective action of 2-group ideal hydrodynamics using the modern formalism of [40–43].[7] This includes the discussion of light degrees of freedom in the hydrodynamic effective action, derivation of the constitutive relations in (4a)-(4c), thermodynamic relations, correlation functions of Noether currents and the spectrum of collective excitations. In section

---

[6]For example, the gradient expansion is discarded outright in [38], and different gradient expansion scheme is applied for spatial and time derivative in [39].

[7]See also [44] for review and [45–49] for alternative recent formulation.

4, we study the holographic model of (12) where we discussed holographic dictionaries for conserved currents and operators in the EFT of Section 3 and confirm hydrodynamic prediction of equilibrium current and speed of sound in (9)-(11). Open questions and interesting future directions are discussed in Section 5.

There are three appendices. In Appendix A, we show that the the coefficients of new terms in the constitutive relation, beyond those in hydrodynamic with $U(1)$ zero-form and $U(1)$ one-form symmetry,[8] are fixed by 2-group structure constants and thermodynamic quantities by demanding that the entropy production vanished at ideal hydrodynamic level. This is analogous to the constraints of anomaly induced transport coefficient of [30]. In Appendix B, we show that the macroscopic modes in MHD effective theory (i.e. those with only $U(1)$ one-form symmetry) is dual to Wilson surface in the bulk as well as how to extract its transformation properties. Appendix C, we note some useful formulae that were used to obtain the constitutive relation from the effective action.

**NOTE ADDED**: Upon completion of the first version of this manuscript, we became aware of [50] which discussed details of holographic dictionary used in Section 4 as well as generalisation to higher-group structure beyond 2-group.

## 2 2-group background field and equilibrium partition function

We will begin by studying aspects of quantum field theories that enjoy a 2-group global symmetry in *thermal equilibrium*. As usual, such thermodynamic aspects can be understood by studying the theory on $\mathbb{R}^3 \times S^1$, where the $S^1$ is the compact Euclidean time direction. It is instructive to study the decomposition of the various symmetry currents under a this dimensional reduction; we will show that this permits a simple way to understand the novel terms appearing in (4c).

### 2.1 1-form symmetries in thermal equilibrium

To orient ourselves and develop some machinery, we begin by studying a simpler system: we first review the description of system with only a single 1-form symmetry in thermal equilibrium, i.e. a system with the conserved 2-form current

$$\nabla_\mu J^{\mu\nu} = 0 \,. \tag{13}$$

As described in detail in [4], this is expected to correspond to the universality class of relativistic magnetohydrodynamics. This statement, however requires some refinement; some associated subtleties have been discussed previously in [5,6], and we here review some of their arguments in slightly different language.

Let us denote the compact $S^1$ direction by $\tau$ and consider how the current $J^{\mu\nu}$ decomposes in the dimensionally reduced theory. On $\mathbb{R}^3$, we now have a 0-form symmetry $U(1)_0$ with current $J^{i\tau}$, and a 1-form symmetry $U(1)_1$ with current $J^{ij}$. Physically, $J^{i\tau}$ is the magnetic field 3-vector; its divergencelessness $\nabla_i J^{i\tau} = 0$ is equivalent to the conservation of magnetic flux. $\epsilon^{ijk} J_{jk}$ is the electric field three-vector; we will see that (to leading order in derivatives) it vanishes in equilibrium.

To understand the thermodynamics, we now need to understand how the currents $J^{i\tau}$, $J^{ij}$ are realized in the dimensionally reduced theory. Different choices for the realization of these symmetries – e.g. spontaneously broken, unbroken, etc. – correspond to different phases of the plasma. Somewhat counter-intuitively, $U(1)_0$ is actually *spontaneously broken* in the "normal

---

[8]See e.g. Section V of [4] and Section II.D. of [43] for more details on the constitutive relation and effective action construction of this theory.

phase", i.e. the phase corresponding to a usual finite-temperature plasma, as previously argued in slightly different language in [5,6].

One way to understand this is to note that the assertion that $J^{i\tau}$ is spontaneously broken is equivalent to the usual statement that the magnetic field in a deconfined plasma is "unscreened". To see this more explicitly, consider inserting two static probe magnetic monopoles into the plasma, separated by a distance $L$. As the magnetic field is unscreened, the magnetic field essentially behaves as though as it is in vacuum, and the two monopoles should experience an interaction potential obeying the usual Coulomb law $L^{-1}$. The field that mediates this power-law interaction must therefore be gapless in the $\mathbb{R}^3$ directions. This field is in fact the Goldstone mode associated with the spontaneous breaking of $J^{i\tau}$.[9]

With this understood, we may now write down an effective action for our system on $S^1 \times \mathbb{R}^3$. Let us denote by $\psi$ the Goldstone mode for the spontaneously broken symmetry above; then under a $\tau$-independent 1-form symmetry transformation by $\Lambda_\mu(x^i)$ in the original theory, we have

$$\psi \to \psi + \Lambda_\tau(x^i). \tag{15}$$

Suppose we now insert a probe magnetic monopole whose worldline wraps the $\tau$ direction, it couples to $\psi$ as $\exp(iq_m\psi(x^i))$, which can be thought of as a vertex operator in the dimensionally reduced theory. Indeed this operator transforms by a phase under (15), and so is charged under $J^{i\tau}$. $\psi$ will now mediate the long-range Coulomb interaction between two such monopoles. In other words, there is a non-vanishing monopole-monopole correlation function at large spatial separation

$$\left\langle \exp(-iq_m\psi(x^i)) \exp(iq_m\psi(y^i)) \right\rangle \neq 0. \tag{16}$$

Of course this non-vanishing order parameter implies the presence of a spontaneously broken symmetry, and is crucial for construcing the effective action. A similar symmetry will play an important role in the 2-group case discussed in the next section.

To write down a Euclidean effective action consistent with the above symmetries, let us define the source for $U(1)_0$ to be

$$b_i \equiv \frac{1}{2}(b_{i\tau} - b_{\tau i}). \tag{17}$$

We see that the combination $\partial_i\psi - b_i$ is an invariant 3-vector. To match with usual hydrodynamic notation, we should denote its norm and direction by $\mu_b$ and $h^i$:

$$\mu_b^2 = (\partial_i\psi - b_i)^2, \qquad h^i = \mu_b^{-1}(\partial_i\psi - b_i). \tag{18}$$

As usual in hydrodynamics, $\mu$ is taken to be zeroth order in derivatives. We can now construct the following thermal effective action on $\mathbb{R}^3 \times S^1$:

$$W[b;\psi] = \int d^3x\, p(\mu_b, \beta), \tag{19}$$

where $p$ is an unconstrained function of two variables. The partition function in thermal equilibrium is $Z[b] = \int [D\psi] \exp(W[b;\psi])$.

---

[9]As an aside, this can also be understood perturbatively from a microscopic description; consider studying 4d Maxwell EM coupled to electrically charged matter on $S^1 \times \mathbb{R}^3$. Integrating out the matter, one finds an effective action for the gauge field of the form

$$S = \int d^3x \left(f_{ij}f^{ij} + (\partial_i a_\tau)^2 + m_D^2 \cos(q_e a_\tau) + \cdots\right). \tag{14}$$

The 3d photon remains massless, while the time component of the photon picks up a mass due to the coupling to charged particles moving around the Euclidean time circle. $m_D$ can be understod as the Debye mass. The Goldstone mode $\psi$ is the magnetic dual of the 3d photon.

One can show, using the above action that the equal-time correlation function of $\exp(i\psi)$ at two spatially separated points exhibit long-ranged order. Now, taking functional derivatives of $W$ we can now obtain the form of the 2-form current in equilibrium:

$$J^{i\tau} = \frac{\delta W}{\delta b_i} = \frac{\partial p}{\partial \mu_b} h_i\,. \tag{20}$$

This is the usual expression for the magnetic flux in relativistic magnetohydrodynamics, here obtained from an action principle. Note that – unlike in conventional hydrodynamics – this Euclidean action has a single gapless (in the $\mathbb{R}^3$ directions) degree of freedom $\psi$ whose equation of motion we must impose; this equation of motion is simply the conservation of magnetic flux:

$$\nabla_i J^{i\tau} = 0\,. \tag{21}$$

If we write the action above in a generally covariant form it is also straightforward to derive the expression for the stress tensor $T^{\mu\nu}$; we do not do so here as we will obtain it from a more sophisticated action principle in Section 3. More details concerning how this light field $\psi$ is connected to those discussed in [5, 6, 43] and how it is encoded in the holographic dual can be found in Appendix B[10]

One could also imagine a phase where $U(1)_0$ is unbroken; this corresponds to the *superconducting* or Higgs phase in thermal equilibrium. There magnetic flux is confined, and so the correlation function of the vertex operators above in (16) should decay exponentially with separation in the $\mathbb{R}^3$ directions. Within the hydro description this simply means that the correlator vanishes. We do not discuss this phase any further in this work.

## 2.2  2-group global symmetries in thermal equilibrium

We now repeat the earlier analysis for the theory with a 2-group global symmetry where we have both a 0-form symmetry with current $j^\mu$ and a 1-form symmetry with current $J^{\mu\nu}$. We now study how all of these symmetry currents decompose if we write the theory on $S^1 \times \mathbb{R}^3$. We will denote these currents and their corresponding symmetries by:

$$J^{i\tau} \qquad U(1)_0^B\,, \tag{22}$$

$$J^{ij} \qquad U(1)_1^B\,, \tag{23}$$

$$j^i \qquad U(1)_0^A\,, \tag{24}$$

$$j^\tau \qquad U(1)_{-1}^A\,. \tag{25}$$

Note that from the point of view of the dimensionally reduced theory, $j^\tau$ can be formally thought of as a "-1-form" current. This is a somewhat formal statement: after all, the "current" for a -1-form symmetry does not obey a conservation law (though its charge does obey a quantization condition: $\int d^3 x\, j^\tau = \mathbb{Z}$), but we will see the utility of thinking in this formal manner later.

Recall now that the external source for the original 0-form current was $a_\mu$. It is now helpful to define:

$$\beta \mu_a \equiv \int_\tau a_\tau \tag{26}$$

as the gauge-invariant Wilson loop of the source around the $\tau$ direction. Note that formally, this is the "source for the $-1$-form current".

---

[10]Note that here we are studying only equilibrium states. If we were studying instead real-time evolution, (the analogue of) $\psi$ has no dynamics, (see Section B), and it does not introduce new light degrees of freedom in the familiar MHD equations.

It is also helpful to record how the sources change under a $\tau$-independent 1-form and 0-form transformation, which follow directly from the dimensional reduction of the corresponding 4d expressions.

$$U(1)_0^B: \qquad b_{i\tau} \to b_{i\tau} + \partial_i \Lambda_\tau \,, \tag{27}$$

$$U(1)_1^B: \qquad b_{ij} \to b_{ij} + \partial_i \Lambda_j - \partial_j \Lambda_i \,, \tag{28}$$

$$U(1)_0^A: \qquad a_i \to a_i + \partial_i \lambda \,, \qquad b_{ij} \to b_{ij} + \kappa F_{ij}\lambda \,, \qquad b_{i\tau} \to b_{i\tau} + \kappa \partial_i \mu_a \lambda \,. \tag{29}$$

The expressions for the currents in terms of the functional derivatives is:

$$J^{i\tau} = \frac{\delta W}{\delta b_i} \,, \qquad J^{ij} = \frac{\delta W}{\delta b_{ij}} \,, \qquad j^\tau = \frac{\delta W}{\delta \mu_a} \,, \qquad j^i = \frac{\delta W}{\delta a_i} \,. \tag{30}$$

The current $j^i$ now enjoys an interesting non-conservation law:

$$\nabla_i j^i_{\text{cons}} = \kappa \nabla_i \mu_a J^{i\tau} + \frac{\kappa}{2} F_{ij} J^{ij} \,. \tag{31}$$

This is a combination of a (3d) 2-group together with a new structure which one is tempted to call a 1-group, as it involves the (gradient of the) source for a -1-form symmetry together with the current for a 0-form symmetry:

We would now like to write down a theory that is as simple as possible of a generalization of that discussed in the previous subsection, i.e. that it permits an adjustable finite density of 1-form charge. For this to happen we require the existence of an invariant chemical potential $\mu_b$ for the 1-form charge. Note however that the previous definition of $\mu_b$ in (18) is no longer invariant under the extended transformation (27) to (29).

The simplest generalization is to consider a symmetry breaking pattern that breaks $U(1)_0^B \times U(1)_0^A$ down to a diagonal subgroup, where the precise embedding of this subgroup depends on the value of $\mu_a$; i.e. we postulate the existence of a field $\psi$ that transforms as:

$$\psi \to \psi + \Lambda_\tau(x^i) + \hat{\kappa}\mu_a \lambda(x^i) \,. \tag{32}$$

With this choice, it is now possible to write down a 3-vector

$$\partial_i \psi - b_i - \hat{\kappa}\mu_a a_i \,, \tag{33}$$

that is invariant under (27) to (29). Following (18) we may now define a 1-form chemical potential and field direction vector:

$$\mu_b^2 = (\partial_i \psi - b_i - \kappa \mu_a a_i)^2 \,, \qquad h^i = \mu_b^{-1}(\partial_i \psi - b_i - \kappa \mu_a a_i) \,. \tag{34}$$

We believe that the choice of symmetry breaking pattern (32) is the simplest choice that still allows the construction of an invariant chemical potential. We can now use these invariants to write down the 2-group generalization of (19):

$$W[a, b] = \int d^3 x \, p(\mu_a, \mu_b, \beta) \,. \tag{35}$$

From this action, one can show that the long-ranged interaction between the probe monopole in this theory remains unscreened as the correlators (16) is nonzero. One may now take functional derivatives to construct the consistent currents (from the zeroth order terms in the action):

$$J^{i\tau} = \frac{\partial p}{\partial \mu_b} h_i \,, \tag{36}$$

$$j^\tau = \frac{\partial p}{\partial \mu_a} + \kappa a_i J^{i\tau} \,, \tag{37}$$

$$j^i = \kappa \mu_a J^{i\tau} \,. \tag{38}$$

This is precisely the structure of currents described in (4c); we see that the interesting phenomenon of a persistent current in the direction of the magnetic field follows directly from the invariance under the 2-group global symmetry.

We have now succeeded in deriving the structure of equilibrium currents from a generating functional. It is straightforward to formulate the theory above in a generally covariant manner and thereby construct the stress tensor as well; the Ward identities then provide the required equations of motion and the full structure of hydrodynamics.

However, we now instead discuss how to obtain a *dynamical* hydrodynamic theory directly from an action principle, where the hydrodynamic equations of motion arise from demanding that an action is stationary.

## 3  Hydrodynamic effective action

The effective action for a non-dissipative hydrodynamic theory can be obtained by writing the partition function as a path integral over light degrees of freedom $\{X^\mu, \varphi_\mu, \phi\}$ with the effective action $W$

$$Z[g_{\mu\nu}, a_{\mu\nu}, b_{\mu\nu}] = \int \mathcal{D}[X]\mathcal{D}[\varphi]\mathcal{D}[\phi] \exp\left(\frac{i}{\hbar} W[g_{\mu\nu}, b_{\mu\nu}, a_\mu, X^\mu, \varphi_\mu, \phi]\right). \tag{39}$$

Here "$\hbar$" controls the strength of fluctuations, both thermal and quantum; we will focus on the case where $\hbar$ is small and employ the saddle point approximation. All of these degrees of freedom depends on the spacetime $x^\mu$ implicitly through an auxilliary space $\sigma^A(x^\mu) = (\sigma^0, \sigma^i)$, where one can think of $\sigma^i(x^\mu)$ as a labeling each infinitesimal fluid elements as it moves through the spacetime and $\sigma^0$ denoting the internal clock of these elements, see e.g. [44] for a review. The first variable $X^\mu(\sigma)$ is a dynamical field which describes the motion of a fluid element labelled by $\sigma^A$. The field $\varphi_\mu(\sigma(x))$ and $\phi(\sigma(x))$ can formally be thought of as Stueckelberg fields of the one-form and zero-form $U(1)$ symmetry. Upon the background field transformation

$$a_\mu \to a_\mu + \partial_\mu \lambda, \qquad b_{\mu\nu} \to b_{\mu\nu} + \hat{\kappa} F_{\mu\nu} \lambda + 2\partial_{[\mu}\Lambda_{\nu]}, \tag{40}$$

the Stueckelberg fields $\{\phi, \varphi_\mu\}$ simultaneously transform as

$$\phi \to \phi - \lambda, \qquad \varphi_\mu \to \varphi_\mu - \Lambda_\mu. \tag{41}$$

Intuitively, one may think of $\phi(\sigma^0, \sigma^i)$ as a phase of the fluid element labeled by $\sigma^i$ at the internal time $\sigma^0$. Similarly, the integral $\int dx^\mu \varphi_\mu$ should be thought of as a phase associated to a dynamical magnetic flux line.

To connect these degrees of freedom to those that appears in the equilibrium partition function construction, one can set $X^t = \sigma^0 = t$ and $X^i = \sigma^i = x^i$ and analytically continue $t$ to the Euclidean time $\tau$. In configuration where nothing depends on $\tau$, the field $\psi$ introduced in section 2.2 is straightforwardly related to the Stueckelberg fields $\phi, \varphi_\mu$. Using the transformation of $\psi$ under the 2-group background field redefinition in (32), it can be expressed in terms of the Stueckelberg fields as:

$$\psi = -\varphi_\tau - \hat{\kappa}\mu_a \phi. \tag{42}$$

In this section however we will work directly with $\phi$ and $\varphi$, which are more suited to the construction of the hydrodynamic effective action.

Let us review a few more advantages of choosing $\{X^\mu, \varphi, \phi\}$ as the hydrodynamic degrees of freedom. To construct a theory invariant under the background field transformation (40),

we demand that the action $W$ depends only on the the combinations of $g, a, b$ and Stueckelberg fields through the following combinations

$$G_{AB} = g_{\mu\nu} \frac{\partial X^\mu}{\partial \sigma^A} \frac{\partial X^\nu}{\partial \sigma^B}, \tag{43a}$$

$$A_A = \frac{\partial X^\mu}{\partial \sigma^A} \left( a_\mu + \partial_\mu \phi \right), \tag{43b}$$

$$B_{AB} = \frac{\partial X^\mu}{\partial \sigma^A} \frac{\partial X^\nu}{\partial \sigma^B} \left( b_{\mu\nu} + 2\partial_{[\mu}\varphi_{\nu]} + \hat{\kappa}(da)_{\mu\nu}\phi \right). \tag{43c}$$

By construction $\{G, A, B\}$ are invariant under the diffeomorphism of $g_{\mu\nu}$ and background gauge transformation of $a_\mu$ and $b_{\mu\nu}$. Thus the effective action $W[G, A, B]$ is guaranteed to be invariant. In addition, the transformation of the Stueckelberg field implies that the Euler Lagrange equation of $X^\mu, \phi$ and $\varphi_\mu$ are nothing but the conservation of energy-momentum, one-form current $j^\mu$ and two-form current $J^{\mu\nu}$.

$$\partial_\mu \langle T^{\mu\nu} \rangle = \frac{1}{2} H^\nu{}_{\rho\sigma} \langle J^{\rho\sigma} \rangle + F^{\nu\rho} \left( \langle j_\rho \rangle + \hat{\kappa} \langle J_{\rho\sigma} \rangle a^\sigma \right), \tag{44a}$$

$$\partial_\mu \langle j^\mu \rangle = \hat{\kappa} F_{\mu\nu} \langle J^{\mu\nu} \rangle, \tag{44b}$$

$$\partial_\mu \langle J^{\mu\nu} \rangle = 0, \tag{44c}$$

where $H = db - \hat{\kappa} a \wedge da$ is an invariant field strength. These Ward identities arise from the symmetry transformation (3). As before, the explicit appearance of the background gauge field might seem alarming but this is purely kinematic and arises from the fact that $\langle j^\mu \rangle$ also transformed under $a \to a + d\lambda$ according to (6).

We further assume that the generating function admitted a gradient expansions in terms of the local variables namely

$$W = \int d^{d+1}x \sqrt{-g} \left( p(G, A, B) + \mathcal{O}(\partial) \right), \tag{45}$$

where $p$ is a local scalar that depends on $\{G_{AB}, A_A, B_{AB}\}$.

Note, however, that not all components of $G_{AB}, A_A$ and $B_{AB}$ can enter the effective action. This is because the action is also required to be invariant with respect to the internal symmetry of the world sheet $\sigma^A$ and the Stueckelberg fields $\phi, \varphi_\mu$. These internal symmetries determined the phase of the hydrodynamical system that the above effective action realised; see e.g. [51–53]. We shall discussed the physical meaning of these internal symmetry and its consequences in Section 3.1.

## 3.1 Internal symmetries of Stueckelberg fields

In this section, we discuss the internal symmetry of the fluid with 2-group global symmetry, which forbid certain components of $G_{AB}, A_A, B_{AB}$ that enters the effective action. As a result of this procedure, one will be able to related components of these quantities to more familiar hydrodynamics variables such as temperature, chemical potential, fluid velocity etc. (4a)-(4c). Moreover, we will show, in Section 4 how to derive the transformation generated by these internal symmetries from holographic dual description.

One demands the following internal symmetry of the world volume coordinate $\sigma^A$

(i) *Spatial relabeling* where one can choose to "rename" the infinitesimal fluid elements $\sigma^i$ at a specific time $\sigma^0 = constant$ by the different label $\sigma'^i$. This gives

$$\sigma^i \to \sigma'^i(\sigma^j). \tag{46a}$$

(ii) *Time-shift* where one allows to choose the initial time for each fluid elements $\sigma^i$. This symmetry is manifested as

$$\sigma^0 \to \sigma^0 + f(\sigma^i). \tag{46b}$$

These world volume symmetries can be thought of as the defining properties of a fluid.[11] Demanding that the effective Lagrangian to be invariant(46) restricts the components of $G_{ab}$ it can depends on. We can see that $G_{0i}$ and $G_{ij}$ are not invariant under the spatial relabeling (46a). The only allowed component is therefore $G_{00}$, which defines the temperature in the following way

$$G_{00} = g_{\mu\nu} \frac{\partial X^\mu}{\partial \sigma^0} \frac{\partial X^\nu}{\partial \sigma^0} \equiv -\frac{1}{T^2}. \tag{47}$$

One can view this quantity as a norm for a vector $\partial X^\mu / \partial \sigma^0$, where the latter is also invariant under internal symmetries of $\sigma^A$. We can then use it to define a unit vector which play the role of the fluid velocity as

$$\frac{\partial X^\mu}{\partial \sigma^0} = \frac{u^\mu}{T}, \qquad u^\mu u_\mu = -1. \tag{48}$$

The next requirement came from the fact that demand that theory has an unbroken zero-form $U(1)$ symmetry. This means that we require the correlation function of two point-like operators charged under the zero-form $U(1)$ to decay exponentially, i.e. have vanishing correlation function at large distance. For the effective action written in Eq.(45), these operators are vertex operators defined as

$$V(x) = \exp(-i\phi(x)), \qquad V^\dagger(x) = \exp(i\phi(x)). \tag{49}$$

A way to demand that equal-time correlation function of these charge operators vanished is to demand that the theory is invariant under the following transformation

(iii) *Zero-form shift* where we have a freedom to arbitrarily assign the phase $\phi$ at initial time $\sigma^0$.

$$\phi \to \phi + c(\sigma^i), \tag{50}$$

which guaranteed that $\langle V^\dagger(t, x^i) V(t, y^i) \rangle = 0$ otherwise it will become multi-valued.[12] This symmetry is the well-known *chemical shift*; if we do not impose this, we find instead the effective theory of a superfluid, see e.g. [45, 54]. In addition, we also demand that the field $\varphi_A = (\partial X^\mu / \partial \sigma^A) \varphi_\mu$ transformed under this shift symmetry as

$$\varphi_A \to \varphi_A - \hat{\kappa} c(\sigma^i) A_A, \tag{51}$$

whose meaning will be elaborated below.

The only component of $A_A$ invariant under (50) is $A_0$, which gives us the zero-form chemical potential

$$A_0 = \frac{\partial X^\mu}{\partial \sigma^0} (a_\mu + \partial_\mu \phi) = \frac{\mu_a}{T}. \tag{52}$$

Lastly, we impose a shift symmetry for the one-form Stueckelberg field $\varphi_\mu$ that is known to produce the correct hydrodynamic description of a magnetohydrodynamics [43].

---

[11] In the construction of [45], the effective action is formulated with the choice of $\sigma^0$ chosen to be $\sigma^0 = X^0$. See also discussion in section V.D. of [40].

[12] More precisely, this correlator decays exponentially in a theory with unbroken zero-form $U(1)$, and is only zero at scales longer than a microscopic correlation length which is taken to be arbitrarily small in the hydro limit.

(iv) *One-form shift* where $\varphi_\mu$ transforms as:

$$\varphi_0 := \frac{\partial X^\mu}{\partial \sigma^0} \varphi_\mu \to \varphi_0\,, \qquad \varphi_i := \frac{\partial X^\mu}{\partial \sigma^i} \varphi_\mu \to \varphi_i + C_i\left(\sigma^j\right). \tag{53}$$

Intuitively, this symmetry force the correlation function of a closed spatial t'Hooft line $W(\gamma) = \mathcal{P} \exp\left(i \int_\gamma d\sigma^i \varphi_i\right)$ to vanish; this is expected in the normal phase of a plasma.

We can demand the invariance under (53) to show that only $B_{0i}$ component is invariant. Note however, that $B_{0i}$ is not invariant under (51) as it transformed as

$$B_{0i} \to B_{0i} + \hat{\kappa}(\partial_i c)A_0\,.$$

Nevertheless, one can combine a vector $B_{0i}$ with the product of $A_0 A_i$ to make an invariant combinations analogous to the construction of effective action for anomalous hydrodynamics [35,55,56]. This allows us to define the one-form chemical potential as

$$\frac{\mu_b h_i}{T} = B_{0i} - \hat{\kappa} A_0 A_i\,,$$

which can be converted from the labeling space $\{\sigma^0, \sigma^i\}$ to a physical spacetime $x^\mu$ as

$$\mu_b h_\mu = u^\nu B_{\nu\mu} - \hat{\kappa}\mu_a A_\mu^\perp\,, \qquad A_\mu^\perp = \Delta_\mu{}^\nu(a_\nu + \partial_\nu \phi)\,, \tag{54}$$

with $\Delta^{\mu\nu} = g^{\mu\nu} + u^\mu u^\nu$. Here, we define $h_\mu$ to be a unit vector, $h^\mu h_\mu = 1$ such that a scalar quantity can be constructed via $\mu_b = \sqrt{g^{\mu\nu}(\mu_b h_\mu)(\mu_b h_\nu)}$.

Let us further elaborate on the zero-form shift of $\varphi_\mu$ in (51). This can be understood as a covariant version of the transformation of $\psi$ postulated in the KK reduced theory of Section 2, using the relations between $\psi$ and $\varphi_\mu, \phi$ alluded to in (42). One may also consider the the equilibrium configuration where nothing depends on $\tau \sim \sigma^0$ and $u^\mu = \delta^{\mu\tau}$, to show that the definition of $\mu_b$ in (54), required by the one-form shift (51), is equivalent to those obtained in equilibrium partition function analysis in Eq. (34).

All in all, the effective Lagrangian after imposing the internal symmetry at zeroth derivative level can be written as

$$\mathcal{L} = p(G_{00}, A_0, B_{0i} - \hat{\kappa} A_0 A_i) = p(T, \mu_a, \mu_b)\,. \tag{55}$$

Upon variation w.r.t. to the background fields, one obtain the one-point function of the Noether currents, namely

$$\langle T^{\mu\nu} \rangle = \frac{2}{\sqrt{-g}} \frac{\delta W}{\delta g_{\mu\nu}}\,, \qquad \langle J^{\mu\nu} \rangle = \frac{2}{\sqrt{-g}} \frac{\delta W}{\delta b_{\mu\nu}}\,, \qquad \langle j^\mu \rangle = \frac{1}{\sqrt{-g}} \frac{\delta W}{\delta a_\mu}\,. \tag{56}$$

With some algebraic manipulation, one obtain the constitutive relation in (4a)-(4c). Some useful formulae aiding derivation of this result can be found in appendix C

## 3.2 Retarded 2-point functions and hydrodynamic modes

In this section, we point of a few retarded correlation functions which encodes interesting hydrodynamic modes. In general, the retarded correlators can be obtained by varying the generating one-point function w.r.t. the background fields in the following way (see e.g. [57]

for a review)

$$\delta\left(\sqrt{-g}\langle T^{\mu\nu}(x)\rangle\right)$$
$$= -\int d^{d+1}y\left[\frac{1}{2}G_{TT}^{\mu\nu,\rho\sigma}(x,y)\delta g_{\rho\sigma}(y) + \frac{1}{2}G_{TJ}^{\mu\nu,\rho\sigma}(x,y)\delta b_{\rho\sigma}(y) + G_{Tj}^{\mu\nu,\alpha}(x,y)\delta a_{\alpha}(y)\right],$$
$$\delta\left(\sqrt{-g}\langle J^{\mu\nu}(x)\rangle\right)$$
$$= -\int d^{d+1}y\left[\frac{1}{2}G_{JT}^{\mu\nu,\rho\sigma}(x,y)\delta g_{\rho\sigma}(y) + \frac{1}{2}G_{JJ}^{\mu\nu,\rho\sigma}(x,y)\delta b_{\rho\sigma}(y) + G_{Jj}^{\mu\nu,\alpha}(x,y)\delta a_{\alpha}(y)\right],$$
$$\delta\left(\sqrt{-g}\langle j^{\mu}(x)\rangle\right)$$
$$= -\int d^{d+1}y\left[\frac{1}{2}G_{jT}^{\mu,\rho\sigma}(x,y)\delta g_{\rho\sigma}(y) + \frac{1}{2}G_{jJ}^{\mu,\rho\sigma}(x,y)\delta b_{\rho\sigma}(y) + G_{jj}^{\mu,\alpha}(x,y)\delta a_{\alpha}(y)\right],$$

with $G_{XY}(x,y)$ denotes retarded correlation functions of operators $X(x)$ and $Y(y)$ and $\delta(\sqrt{-g}X)$ on the l.h.s. means the difference between one-point function with and without background fields perturbations. Operationally, it is amount to solving the perturbed thermodynamic quantities e.g. $\{\delta T, \delta\mu_a, \delta\mu_b\}$ and the $\{\delta u^{\mu}, \delta h^{\mu}\}$ in terms of $\{\delta g, \delta b, \delta a\}$, plug it back into the constitutive relations and then take the variational derivative with the appropriate background fields.

We focus on the response function of the equilibrium configuration with the string-like objects are aligned parallel to the $z$−direction i.e. $\langle J^{tz}\rangle = \rho_b$ for simplicity. The clearest signature of 2-group occur in the (Fourier transformed) correlators $G_{XY}(\omega, \mathbf{q})$ with $q_x = q_y = 0$ so we will restrict our presentation to this configuration as well.

There are two decoupled fluctuations channel: those that are transverse and perpendicular to $\mathbf{q}$. The fluctuations in the transverse channel is simple, as it only involves fluctuations of $\delta u^{\perp}, \delta h^{\perp}$ with $\perp = x, y$ in our setup. The resulting retarded correlation functions are

$$G_{TT}^{t\perp,t\perp}(\omega, q_z) = \frac{1}{\mathcal{P}_{\perp}(\omega, q_z)}\left[(\omega\varepsilon + \hat{\kappa}\mu_a^2\rho_b q_z)^2 + p(\omega^2\varepsilon + \mu_b\rho_b q_z^2)\right],$$
$$G_{TJ}^{t\perp,t\perp}(w, q_z) = -\frac{\rho_b q_z}{\mathcal{P}_{\perp}(\omega, q_z)}\left[\omega(\varepsilon + p) + \hat{\kappa}\mu_a^2\rho_b q_z\right], \tag{57}$$
$$G_{JJ}^{t\perp,t\perp}(\omega, q_z) = -\frac{(\rho_b q_z)^2}{\mathcal{P}_{\perp}(\omega, q_z)},$$

with $\mathcal{P}_{\perp}(\omega, q_z)$ is the polynomial encoding the collective modes responsible for transporting the transverse momentum $T^{t\perp}$ and transverse fluctuation of the string $J^{t\perp}$. Its explicit form is

$$\mathcal{P}_{\perp}(\omega, q_z) = (\varepsilon + p)\omega^2 + 2\hat{\kappa}\mu_a^2\rho_b\omega q_z - \mu_b\rho_b q_z^2, \tag{58}$$

which resulting in the following spectrum

$$\omega_{\perp} = \left(-\frac{\hat{\kappa}\mu_a^2\rho_b}{\varepsilon + p} \pm \sqrt{\mathcal{V}_A^2 + \left(\frac{\hat{\kappa}\mu_a^2\rho_b}{\varepsilon + p}\right)^2}\right)q_z, \qquad \mathcal{V}_A^2 = \frac{\mu_b\rho_b}{\varepsilon + p}. \tag{59}$$

The longitudinal fluctuations is more complicated. There are three independent variables one needed to solved namely $\{\delta T, \delta\mu_a, \delta u^z\}$ and the pole of the retarded correlators is governed by a cubic equation in $\omega$ for a generic equations of states.[13] To get better understanding, and to disentangle the effect of nonzero 2-group structure constant $\hat{\kappa}$, one can study the limit

---

[13]In the case where $\hat{\kappa} = 0$ or $\rho_b = 0$, one of the root of such equation is $\omega = 0$ and the others two are typical sound modes $\omega \propto \pm q_z$. This spectrum is the same as in fluid with finite ordinary $U(1)$ charge density.

of zero charge density and chemical potential. This amounts to setting $\rho_a = 0, \mu_a = 0$ in equilibrium and write the fluctuation of density as $\delta\rho_a = \chi_{aa}\delta\mu_a$ where $\chi_{aa}$ is the (zero-form) charge susceptibility. In this limit, the zero-form charge fluctuations decouple from the momentum and energy and the resulting correlation functions involving $j^\mu$ are

$$
G_{jj}^{t,t}(\omega, q_z) = -\frac{2\hat{\kappa}\rho_b q_z}{\omega + (2\hat{\kappa}\rho_b/\chi_{aa})q_z}, \qquad G_{jj}^{z,z}(\omega, q_z) = \frac{(2\hat{\kappa}\rho_b)^2 \omega}{\chi_{aa}(\omega + (2\hat{\kappa}\rho_b/\chi_{aa})q_z)},
$$
$$
G_{jj}^{t,z}(\omega, q_z) = G_{jj}^{z,t}(\omega, q_z) = -\frac{\hat{\kappa}\rho_b(\omega\chi_{aa} - 2\hat{\kappa}\rho_b q_z)}{\omega\chi_{aa} + 2\hat{\kappa}\rho_b q_z}.
\tag{60}
$$

where one find a chiral propagating mode with the spectrum governing the poles

$$
\omega = -\frac{2\hat{\kappa}\rho_b}{\chi_{aa}} q_z.
\tag{61}
$$

These are the same correlation functions and modes one finds in the charge fluctuations of an ideal fluid with chiral anomaly in $1 + 1$ dimensions[14] (for example at the edge of of quantum hall systems) with anomaly coefficients replaced by $\hat{\kappa}\rho_b$. This is not surprising since 2-group theory reduced on spatial $T^{d-1}$ orthogonal to the strings has the same anomalous Ward identity as the $d + 1 = 2$ chiral fermion [19]. The computation for $G_{jj}^{\mu,\nu}(\omega, q_z)$ can be done easily in the holographic context and we confirm the existence of these chiral sound mode as well as hydrodynamic prediction of the chiral propagating mode (61) in Section 4.

# 4 Minimalist' holographic dual

In this section we study a simple holographic dual of a theory with 2-group global symmetry. This theory was first discussed in [19]. The dynamical bulk fields are a one-form $\mathcal{A}_a(X)$ and $\mathcal{B}_{ab}(X)$, where $X^a = \{x^\mu, r\}$ is the bulk coordinate.

The bulk action is:

$$
S_{bulk} = -\int d^{d+2}X \sqrt{-G}\left(\frac{1}{4}\mathcal{F}_{ab}\mathcal{F}_{ab} + \frac{1}{6}\mathcal{H}_{abc}\mathcal{H}^{abc}\right),
$$

where $\mathcal{H} = d\mathcal{B} - \hat{\kappa}\mathcal{A}\wedge d\mathcal{A}$. The transformation of these bulk fields under the 2-group symmetry is as in (3) but with the transformation parameters $\{\lambda(x^\mu, r), \Lambda_a(x^\mu, r)\}$ depending on the holographic coordinates. As usual in holography, we identify two fields at the boundary to be the source $a_\mu, b_{\mu\nu}$

$$
\mathcal{A}_\mu(r \to \infty) = a_\mu, \qquad \mathcal{B}_{\mu\nu}(r \to \infty) \sim b_{\mu\nu},
\tag{62}
$$

Some relevant details on how to perform holographic renormalisation involving the 3-form field strength for action of this type can be found in [58–60]; it requires a double trace-type deformation at the boundary.

We now construct the holographic dictionary. The one-point function can be found by varying the action with respect to $a_\mu, b_{\mu\nu}$ at the boundary, resulting in

$$
\langle j^\mu \rangle = -\sqrt{-G}(\mathcal{F}^{r\mu} + 2\hat{\kappa}\mathcal{H}^{r\mu\nu}\mathcal{A}_\nu)\Big|_{r\to\infty} = -\sqrt{-G}\mathcal{F}^{r\mu}(r \to \infty) + \hat{\kappa}J^{\mu\nu}a_\nu,
$$
$$
\langle J^{\mu\nu} \rangle = -2\sqrt{-G}\mathcal{H}^{r\mu\nu}\Big|_{r\to\infty}.
\tag{63}
$$

This implies that the field strength alone is the *covariant* current, defined in (7),

$$
j_{cov}^\mu = -\sqrt{-G}\mathcal{F}^{r\mu}(r \to \infty).
\tag{64}
$$

---

[14]See e.g. [35] for a nice recent discussion of this kind of transport.

Note also that the $r$−component for the equations of motion for the gauge field $\mathcal{A}_a$ becomes the Ward identity of the convariant current at the boundary i.e.

$$\nabla_a \mathcal{F}^{ab} = -\hat{\kappa}(2\mathcal{H}^{bac})\mathcal{F}_{ac} \qquad \Rightarrow \qquad \partial_\mu \langle j^\mu_{cov}\rangle = \hat{\kappa}\langle J^{\mu\nu}\rangle (da)_{\mu\nu}. \tag{65}$$

The purpose of this section is two fold. First, we perform a conceptual exercise: we show how the macroscopic degrees of freedom $\phi, \varphi_\mu$ discussed in Section 3 are encoded in this holographic model. Moreover, we argue that the action has to depend on the combination $A_\mu, B_{\mu\nu}$ in (43), and thus that one can derive from holography the somewhat peculiar structure of the chemical shift symmetry in (50)-(51) and (53). (The deconstruction of $G_{AB}$ from holography at ideal limit can be found in [61] and we will not focus on it in this section.)

Second, we perform a more conventional holographic calculation to verify some predictions of our hydrodynamic theory: we show that this model exhibits the equilibrium current along the magnetic field line in (9) and a chiral propagating mode in (11).

## 4.1 Holographic deconstruction of 2-group gauge theory

The structure of effective action can be obtained by keeping track of the radial components $\mathcal{A}_r, \mathcal{B}_{\mu r}$. More precisely, the low energy hydrodynamic fields $\phi$ and $\varphi_\mu$ are related to the following bulk line operators:

$$\phi(x^\mu) = \int_{r_h}^\infty dr' \mathcal{A}_r, \qquad \varphi_\mu(x^\mu) = \int_{r_h}^\infty dr' \left(\mathcal{B}_{r\mu} - \phi\, \mathcal{F}_{r\mu}\right), \tag{66}$$

where we see that they transform as a Stueckelberg field of zero-form and one-form parts of the 2-group in (40)-(41) if we perform the large bulk gauge transformation with parameters $\lambda(r, x^\mu)$ and $\Lambda_a = (\Lambda_\mu(r, x^\nu), 0))$ such that

$$\int_{r_h}^\infty dr' \partial_{r'}\lambda(r', x^\mu) = \lambda(\infty, x^\mu), \qquad \int_{r_h}^\infty dr' \partial_{r'}\Lambda_\mu(r', x^\nu) = \Lambda_\mu(\infty, x^\nu). \tag{67}$$

This connection between the bulk Wilson line to the operator in hydrodynamic effective action was explored in [62]. More systematic ways of extracting the action of real-time dynamics at finite temperature (see e.g. [63, 64]) get further developed in [65, 66].

One could use presumably use this formalism to derive the full effective action; we will not do this, and instead will just use it to show that the dual theory must only depends on combinations of $\phi, \varphi_\mu$ and $\mathcal{A}_\mu, \mathcal{B}_{\mu\nu}$ at $r \to \infty$ presented in Section 3. More precisely, the invariance of the effective action, $W$, under the gauge transformation implies that it can only depends on the combination $A_A, B_{AB}$ in (43). Further demanding that the correlation functions of $\phi, \varphi_\mu$ to be independent under a particular set of *residual* gauge transformation imposes the "chemical shift" symmetries and further restricts $W = \int d^{d+1}x \sqrt{-g}\, p$ to those in (55).

This parallel between procedures in the hydrodynamic effective action construction and what happens in the bulk 2-group gauge theory is illustrated in Figure 2.

To start with, the holographic action (12) is gauge invariant. Thus, we can choose a gauge such that the structures in (43) is manifested. This procedure is well-understood for the holographic dual of zero-form $U(1)$ [65]; here we generalize it slightly to the 1-form case and for the 2-group structure. First, we pick the gauge choice such that $\mathcal{A}_a \to \mathcal{A}_a^{(1)}$ with $\mathcal{A}_r^{(1)} = 0$. This gives

$$\mathcal{A}_\mu^{(1)}(r, x^\nu) = \mathcal{A}_\mu(r, x^\nu) + \partial_\mu \phi(r, x^\nu), \qquad \phi(r, x^\mu) = \int_r^\infty dr' \mathcal{A}_r(r, x^\mu). \tag{68}$$

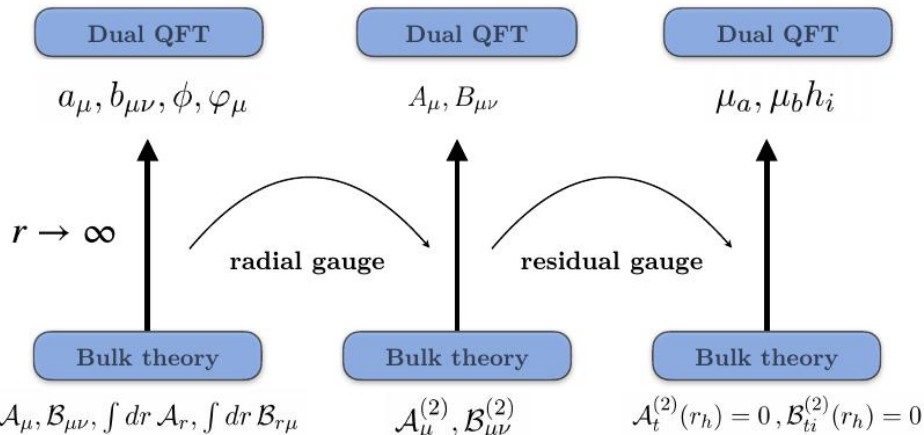

Figure 2: Summary of the procedure which reduced the dependence of bulk fields $\mathcal{A}_a, \mathcal{B}_{ab}$ to thermodynamic quantities $\{\mu_a, \mu_b h_i\}$. From left to right, we list all the bulk fields and the Stueckelberg fields in the dual QFT. Upon imposing the radial gauge, the bulk theory can only depends on $\mathcal{A}_\mu^{(2)}, \mathcal{B}_{\mu\nu}^{(2)}$ in (68),(70) dual to $A_\mu, B_{\mu\nu}$ in (43). Lastly, we identify the residual gauge transformation as the shift symmetry in (50)-(51) and (53). Requiring that the physical quantities is independent of the residual gauge transformation, we conclude that the effective action of the dual QFT can only depends on $\mu_a$ and $\mu_b h_\mu$ in (52) and (54).

The bulk 2-form gauge field becomes

$$\begin{aligned}
\mathcal{B}_{\mu\nu}^{(1)}(r, x^\lambda) &= \mathcal{B}_{\mu\nu}(r, x^\lambda) + \hat{\kappa}\phi(r, x^\lambda)\mathcal{F}_{\mu\nu}(r, x^\lambda), \\
\mathcal{B}_{r\mu}^{(1)}(r, x^\mu) &= \mathcal{B}_{r\mu} + \hat{\kappa}\phi\,\mathcal{F}_{r\mu}(r, x^\nu).
\end{aligned} \tag{69}$$

Next, we perform the one-form $U(1)$ gauge transformation to impose the radial gauge $\mathcal{B}_{ab}^{(1)} \to \mathcal{B}_{ab}^{(2)}$ with $\mathcal{B}_{r\mu}^{(2)} = 0$. This can be done via choosing the transformation parameter $\Lambda_a = (\varphi_\mu(r, x^\nu), 0)$ such that

$$\mathcal{B}_{\mu\nu}^{(2)}(r, x^\lambda) = \mathcal{B}(r, x^\mu) + \hat{\kappa}\phi(r, x^\lambda)\mathcal{F}_{\mu\nu}(r, x^\lambda) + 2\partial_{[\mu}\varphi_{\nu]}(r, x^\lambda), \tag{70}$$

with

$$\varphi_\mu(r, x^\nu) = -\int_r^\infty \left( \mathcal{B}_{r\mu}(r, x^\nu) + \phi\,\mathcal{F}_{r\mu}(r, x^\nu) \right), \tag{71}$$

whereas $\mathcal{A}_a^{(2)} = \mathcal{A}_a^{(1)}$. Near the boundary $r \to \infty$ the bulk operators in (68) and (70) becomes the operator in the dual EFT action defined in (66). Note that the sequence of the gauge choice applied above is chosen since, if one were to fix $\mathcal{B}_{r\mu} = 0$ first, the zero-form $U(1)$ gauge transformation does not preserve the radial gauge for $\mathcal{B}$. The onshell holographic action thus only depends on the near boundary values of the radial gauge bulk fields $\mathcal{A}_\mu^{(2)}, \mathcal{B}_{\mu\nu}^{(2)}$ which becomes $A_\mu, B_{\mu\nu}$ in (43) that enter the effective action.

Next, we shall elaborate on the residual gauge transformation and its connection to the chemical shift. Let us first focus on $\mathcal{A}_\mu^{(2)}$ and consider such transformation which preserve the radial gauge and regularity in the euclidean bulk geometry, namely

$$\mathcal{A}_r^{(2)}(r, x^\mu) = 0, \qquad \mathcal{A}_\tau^{(2)}(r_h, x^\mu) = 0. \tag{72}$$

The residual zero-form gauge transformation that preserve these two conditions is a generated by a shift in $\phi$ by a parameter $c = c(x^i)$ via

$$\phi \to \phi + c(x^i).$$

This is precisely the zero-form shift introduce in (50). The only zeroth derivative quantity invariant under the above transformation is $\mathcal{A}^{(2)}_\tau$ evaluated at the boundary $r \to \infty$. This is the candidate for the chemical potential $\mu_a$ defined in (52).

Let us now turn our attentention to the components of $\mathcal{B}^{(2)}_{\mu\nu}$. We first look for the residual one-form $U(1)$ gauge transformation that preserved the radial gauge $\mathcal{B}^{(2)}_{r\mu} = 0$ and the regularity $\mathcal{B}^{(2)}_{\tau i}(r_h) = 0$. This is transformation is the same as the one-form shift in (53):

$$\varphi_0 \to \varphi_0, \qquad \varphi_i \to \varphi_i + C_i(x^j).$$

This implies that only $\mathcal{B}^{(2)}_{\tau i}(r \to \infty)$ will enters the dual QFT's effective action. This constraint occurs even in the normal MHD case when $\hat{\kappa} = 0$; see Appendix B for more details.

However, for non-zero $\hat{\kappa}$ this is not the end of the story. Under the zero-form residual gauge transformation, we can see that the field $\varphi_\mu$ transformed as in (51)

$$\varphi_\mu \to \varphi_\mu - \hat{\kappa}\mathcal{A}^{(2)}_\mu c(x^i), \tag{73}$$

by using the definition of the $\varphi_\mu$ in Eq. (66). While the zero-form shift for $\varphi_\mu$ and $\phi$ preserve radial gauge condition $\mathcal{B}^{(2)}_{r\mu} = 0$ for all $r$, we can see that the thermodynamic data cannot solely depends on $\mathcal{B}_{\tau i}$ as the latter is not invariant under the residual gauge transformation. The only invariant candidate for the one-form chemical potential is therefore

$$\mathcal{B}^{(2)}_{\tau i}(r, x^\mu) + \hat{\kappa}\mathcal{A}^{(2)}_\tau(r, x^\mu)\mathcal{A}^{(2)}_i(r, x^\mu), \tag{74}$$

which, at the boundary, this combination is precisely the definition of the one-form chemical potential in (54). To conclude, we find the dual of the Stueckelberg fields as well as their transformation properties in accordance with those postulated in Section 3.1. Demanding that the resulting on-shell gravity action at zeroth order in the derivative expansion is gauge invariant, it can only depends on the thermodynamic data $\mu_a, \mu_b$, defined in (52) and (54) respectively.

We should point out that in the above analysis of the residual gauge transformation, we imagined a bulk Euclidean geometry. It is sufficient to analytically continued the Euclidean time $\tau$ to the Lorentzian time $t$ and the equilibrium one-point function $\langle T^{\mu\nu}\rangle, \langle J^{\mu\nu}\rangle, \langle j^\mu\rangle$ becomes those of ideal hydrodynamics. Nevertheless, one also reach the same conclusion by considering bulk dual of the thermal state real-time dynamics, such as those in [63]. This geometry is constructed by sewing together one Euclidean and two Lorentzian bulk in such a way that it corresponds to the Schwinger-Keldysh time contour, see Figure 3. The continuity condition $\mathcal{A}_\tau = \mathcal{A}_t$ at the point where the Euclidean and Lorentzian pieces are glued together allows us to extend regularity condition to the real-time evolution of $\mathcal{A}_t$. These computation can be carry out in the same way as those demonstrated for a holographic dual of the zero-form $U(1)$ in [66] and it does not change our conclusion at zeroth derivative level.

## 4.2 Equilibrium current and chiral sound modes from holography

After a less conventional analysis in the previous section, here we perform standard holographic computations for current one-point and two-point function. We then show that the holographic model exhibit equilibrium currents along the strings' direction as well as propagating chiral mode as promised in Section 1.1.

Before diving into the computation, let us summarise our setup and stating the results. To simplify the computations, we focus on the 'probe limit' where the metric fluctuations are decoupled from of the gauge fields $\mathcal{A}, \mathcal{B}$ and the geometry is fixed as $AdS_5$ Schwarzschild. This corresponds to the scenario where $T$ and $u^\mu = (1, \mathbf{0})$ is fixed in the hydrodynamic description.

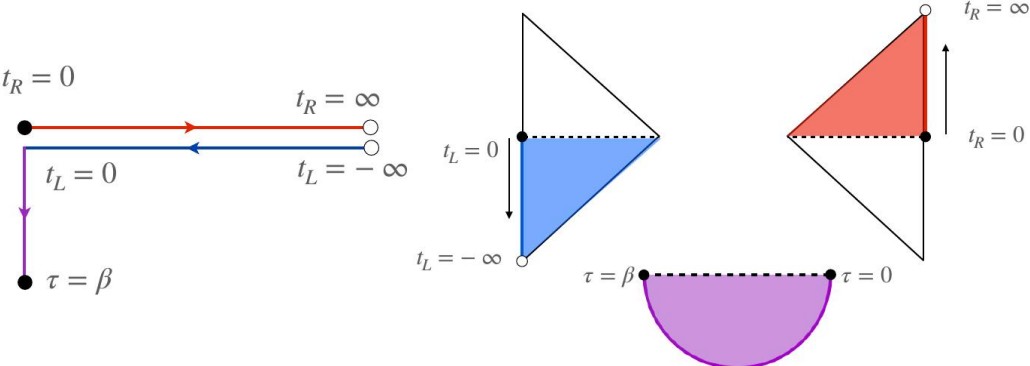

Figure 3: Illustration of the Schwinger-Keldysh time contour for real-time evolution of the thermal state and its corresponding geometry constructing with the procedure in [63, 64], see also [66] for more explicit computations. The bulk geometry is obtained by sewing the two Lorentzian AdS spaces at $t_L = t_R = 0$ with the Euclidean AdS at $\tau = 0$ and $\tau = \beta$ along the dashed line and at $t_R = -t_L = \infty$. This way, the time evolution of the boundary of the space on the right corresponds to the time contour on the left.

The necessary computation is amount to solve for the profile of the following perturbed gauge fields ansatz

$$
\begin{aligned}
\mathcal{A} &= \bar{\mathcal{A}}_t(r)dt + \bar{\mathcal{A}}_z(r)dz + \delta\mathcal{A}_a(t,z,r)dX^a\,, \\
\mathcal{B} &= \bar{\mathcal{B}}(r)dt \wedge dz + \delta\mathcal{B}_{ab}(t,z,r)dX^a \wedge dX^b\,,
\end{aligned}
\tag{75}
$$

on top of the fixed background geometry $ds^2 = r^2(-dt^2 + dx_i dx^i) + dr^2/(r^2 f)$ with $f = 1 - r_h^4/r^4$. As usual, this probe limit where we neglect the backreaction of the gauge fields on the geometry corresponds to the hydrodynamic limit where $T \gg \mu_i, \mu$ so that the charge sector's contribution to the free energy is subleading compared to that of other uncharged degrees of freedom. This limit can also be systematically achieved by demanding that the gauge field sector of the action in (62) is suppressed by a small parameter (e.g. $N_f/N_c$ in the study of probe branes) when added to the Einstein-Hilbert action. This probe limit does not allow us to study 2-group hydrodynamics in the most general equations of state, but will nevertheless highlight the physics of interest. It would be interesting to study the situation including backreaction: this will likely require numerics already at the level of the background, as in [67].

Here, the time-dependent part of $\mathcal{B}$ indicates that the strings in equilibrium configurations is aligned along $z$−direction as in Section 3.2. We find that the equilibrium 1-point function of the covariant current can be obtained by solving for $\bar{\mathcal{A}}_t(r), \bar{\mathcal{A}}_z(r)$ which results in

$$
j^t_{cov} = \chi_{aa}(\rho_b, T)\mu_a\,, \qquad j^z_{cov} = -2\hat{\kappa}\rho_b\mu_a\,,
\tag{76}
$$

where $\chi_a(\rho_b, T)$ can be thought of as charge susceptibility. Its explicit form in $AdS_5$ geometry can be written as

$$
\frac{\chi_{aa}}{2(\pi T)^2} = \frac{2\Gamma\left(\frac{5}{4} - \frac{1}{4}\sqrt{1-(2\mathcal{K})^2}\right)\Gamma\left(\frac{5}{4} + \frac{1}{4}\sqrt{1-(2\mathcal{K})^2}\right)}{\Gamma\left(\frac{3}{4} - \frac{1}{4}\sqrt{1-(2\mathcal{K})^2}\right)\Gamma\left(\frac{3}{4} + \frac{1}{4}\sqrt{1-(2\mathcal{K})^2}\right)}\,,
\tag{77}
$$

with the dimensionless parameter $\mathcal{K} := \hat{\kappa}\rho_b/r_h^2$ with the Hawking temperature $T = r_h/\pi$.

As for the spectrum of fluctuations, we solve for $\delta\mathcal{A}_a, \delta\mathcal{B}_{ab}$. We expect from the hydrodynamic prediction that the chiral propagating mode has speed

$$
c_s = \frac{2\hat{\kappa}\rho_b}{\chi_{aa}} = \frac{\mathcal{K}}{\chi_{aa}/\left(2(\pi T)^2\right)}\,.
\tag{78}
$$

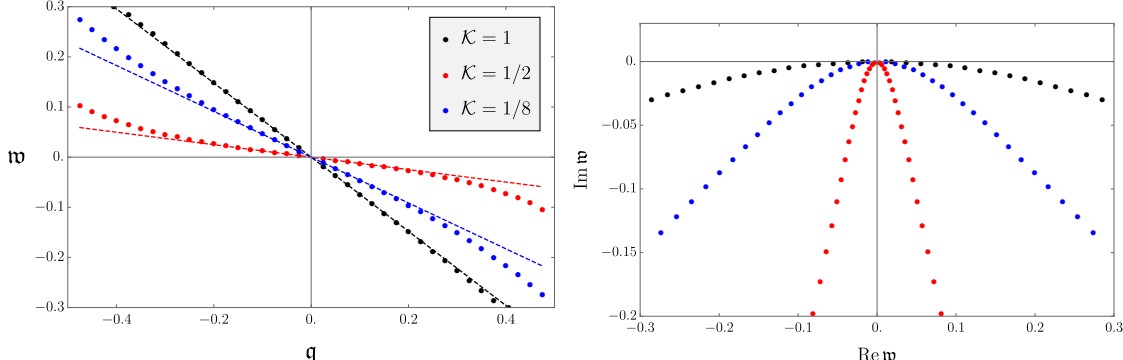

Figure 4: **(LEFT)** The real part of $\mathfrak{w} = \omega/(2\pi T)$ as a function of $\mathfrak{q} = q_z/(2\pi T)$. The • denotes the numerical result while the dashed line corresponds to the predicted dispersion relation in (11) $\operatorname{Re}\omega = -c_s q_z$ with the value of $c_s$ obtained via (79) for $\mathcal{K} = \{1, 1/2, 1/8\}$. **(RIGHT)** The numerical result of quasinormal mode in complex $\mathfrak{w}$−plane showing the behaviour $\operatorname{Im}\mathfrak{w} \sim \mathfrak{q}^2$ at small $\operatorname{Re}\mathfrak{w}$ and/or $\mathfrak{q}$ as we vary **q** from positive to negative values.

To see the mode with this property, we extract the quasinormal mode of the holographic using standard method, such as those in [68]. We find that the quasinormal mode close to the origin in the complex $\omega$ plane is described by

$$\omega = -c_s q_z - i\Gamma_s(q_z)^2\,, \tag{79}$$

where the location of the pole in complex $\omega$ plane as well as the comparison between numerically extracted dispersion relation and the hydrodynamic prediction in Figure 4. The parameter $\Gamma_s$ parametrised the sound attenuation which, while interesting, is beyond the scope of this work. Below, we show some key steps that allows us to obtain these results.

### 4.2.1 Current one-point function

Let us consider the equations of motion for the bulk fields $\mathcal{A}, \mathcal{B}$

$$\nabla_a \mathcal{F}^{ab} + 2\hat{\kappa}\mathcal{H}^{bcd}\mathcal{F}_{cd} = 0\,, \qquad \nabla_a \mathcal{H}^{abc} = 0\,. \tag{80}$$

To find a homogeneous and time-independent solution, we first integrate the equilibrium equations of motion once to arrive at the following first order form:

$$
\begin{aligned}
-2\sqrt{-G}\,\bar{\mathcal{H}}^{rtz} &= \rho_b\,, \\
\sqrt{-G}\,\bar{\mathcal{F}}^{rt} + 2\hat{\kappa}(2\mathcal{H}^{trz})\bar{\mathcal{A}}_z &= Q^t\,, \\
\sqrt{-G}\,\bar{\mathcal{F}}^{rz} + 2\hat{\kappa}(2\mathcal{H}^{zrt})\bar{\mathcal{A}}_t &= Q^z\,,
\end{aligned}
\tag{81}
$$

where $\sqrt{-G}\,\bar{\mathcal{F}}^{rt} = -r^3(\bar{\mathcal{A}}_t)'$, $\sqrt{-G}\,\bar{\mathcal{F}}^{rz} = r^3 f(r)(\bar{\mathcal{A}}_z)'$ with $(...)'$ means derivative w.r.t. the radial coordinate $r$. The parameters $\rho_b, Q^t, Q^z$ are constants of the motion. The horizon regularity conditions $\bar{\mathcal{A}}_t(r_h) = 0$ and $\sqrt{-G}$ and regularity of $\mathcal{A}_z'(r_h)$ (which implies $\bar{\mathcal{F}}^{rz}(r_h) = 0$) together imply that $Q^z = 0$.

Already at this point, we can see that $\bar{\mathcal{F}}_{rz}$ cannot be zero if the chemical potential $\bar{\mathcal{A}}_t(r \to \infty) \neq 0$. This is an indication that equilibrium (covariant) current is, enforced by the horizon regularity, non vanishing.

To obtain the constitutive relation for the covariant current, we use (63)-(64) and find that

$$\langle J^{tz} \rangle = -2\sqrt{-G}\mathcal{H}^{rtz}\Big|_{r\to\infty} = \rho_b\,,$$

$$j^t_{cov} = -\sqrt{-G}\mathcal{F}^{rt}\Big|_{r\to\infty} = r^3(\mathcal{A}^{(0)}_t)'\,,$$

$$j^z_{cov} = -\sqrt{-G}\mathcal{F}^{rz}\Big|_{r\to\infty} = -r^3 f(r)(\mathcal{A}^{(0)}_z)'\,.$$

(82)

Using (81), evaluated at the boundary, we find that

$$\langle j^z_{cov} \rangle = -2\hat{\kappa}\mu_a\rho_b\,,$$

(83)

indicating the presence of the equilibrium current. The expression for $j^t_{cov}$ is slightly more complicated. To do this, we combine (81) into a single decoupled radial evolution for $\bar{\mathcal{A}}_t$, namely

$$\frac{d^2}{du^2}\bar{\mathcal{A}}_t - \frac{1}{f}\left(\frac{\hat{\kappa}\rho_b}{(\pi T)^2}\right)^2\bar{\mathcal{A}}_t = 0\,,$$

(84)

where we introduce the normalised radial coordinate $u = (r_h/r)^2$. The above equation can be easily solved and, upon imposing regularity condition at the horizon $u = 1$, one finds

$$\bar{\mathcal{A}}_t(u) = \mu_a\left[{}_2F_1\left(\alpha,\beta;\frac{1}{2};u\right) - (\chi_a u)\,{}_2F_1\left(-\beta,-\alpha;\frac{3}{2};u\right)\right]$$

$$= \mu_a - \left(\frac{\chi_{aa}\mu_a}{2(\pi T)^2}\right)u + \mathcal{O}(u^2)\,, \qquad \text{near } u\to 0\,,$$

(85)

where parameters in the above expressions are

$$\alpha = -\frac{1}{4}\left(1 + \sqrt{1-(2\mathcal{K})^2}\right)\,, \qquad \beta = \frac{1}{2} - \alpha\,, \qquad \mathcal{K} = \frac{\hat{\kappa}\rho_b}{(\pi T)^2}\,,$$

(86)

and $\chi_{aa}$ is expressed in (77). Plugging the asymptotic solution for $\bar{\mathcal{A}}_t$ into the holographic expression for $j^\mu_{cov}$ in (64), one finds that it becomes the constitutive relation in (76).

### 4.2.2 Quasinormal modes

The spectrum of perturbations can be obtained by solving the linearised perturbation equations for $\delta\mathcal{A}_a, \delta\mathcal{B}_{ab}$. It turns out that the relevant perturbation for zero-form charge fluctuations only involves $\delta\mathcal{A}_t, \delta\mathcal{A}_z$ and $\delta\mathcal{A}_r$. Their linearised equations of motion in the Fourier space $(w, q_z)$ can be written as

$$\left(r^3(\delta\mathcal{A}'_t + i\omega\delta\mathcal{A}_r) - 2\hat{\kappa}\rho_b\delta\mathcal{A}_z\right)' - 2i\hat{\kappa}\rho_b q\delta\mathcal{A}_r + \frac{q_z}{rf}\left(\omega\delta\mathcal{A}_z + q_z\delta\mathcal{A}_t\right) = 0\,,$$

$$\left(r^3 f(\delta\mathcal{A}'_z - iq_z\delta\mathcal{A}_r) - 2\hat{\kappa}\rho_b\delta\mathcal{A}_t\right)' + 2i\hat{\kappa}\rho_b\omega\delta\mathcal{A}_r + \frac{\omega}{rf}\left(\omega\delta\mathcal{A}_z + q_z\delta\mathcal{A}_t\right) = 0\,,$$

$$\omega\left(r^3\delta\mathcal{A}'_t - 2\hat{\kappa}\rho_b\delta\mathcal{A}_z\right) + q\left(r^3 f\delta\mathcal{A}'_z - 2\hat{\kappa}\rho_b\delta\mathcal{A}_t\right) - ir^3\left(\omega^2 - q_z^2 f\right)\delta\mathcal{A}_r = 0\,,$$

(87)

where the last first order equation becomes the Ward identity for $j^\mu_{cov}$ when evaluated at the boundary. Note that the equation of motion in the radial gauge is almost identical to those found in Maxwell-Chern-Simons theory in asymptotic $AdS_3$ geometry (see e.g. [69]) up to the form of $f$ and power of $r$. Unlike that case, however we study this theory using the usual AdS/CFT boundary conditions.

To solve this system of equations we consider the gauge invariant mode

$$Z = \mathfrak{w}\delta\mathcal{A}_z + \mathfrak{q}\delta\mathcal{A}_t\,, \qquad \mathfrak{w} = \frac{\omega}{2\pi T}\,, \qquad \mathfrak{q} = \frac{q_z}{2\pi T}\,.$$

(88)

Its equation of motion is best written in the $u = (r_h/r)^2$ coordinate ranging from $u \in [0,1)$, which can be explicitly written as

$$Z''(u) + \left( \frac{\mathfrak{w}^2 f'(u)}{f(u)(\mathfrak{w}^2 - \mathfrak{q}^2 f(u))} \right) Z'(u) + \frac{1}{u f(u)^2} \left( \mathfrak{w}^2 - \left( \mathfrak{q}^2 + u \mathcal{K}^2 - \frac{\mathcal{K} \mathfrak{w} \mathfrak{q} f'(u)}{\mathfrak{w}^2 - \mathfrak{q}^2 f(u)} \right) f \right) Z(u) = 0 \,, \tag{89}$$

with the $(\ldots)'$ in the above equation now denotes the derivative in $u$. The quasinormal mode can be obtained using Frobenius method as in [68] where we expressed $Z$ as

$$Z(u) = (1-u)^{-i\mathfrak{w}/2}(1+u)^{-\mathfrak{w}/2} \sum_{n=1}^{N} c_n (1-u)^n \,, \tag{90}$$

and the Dirichlet boundary is imposed by

$$Z(u=0) = 0 = \sum_{n=1}^{N} c_n \,. \tag{91}$$

For our purpose, $N = 20$ is enough to reproduce the first three decimal places of quasinormal mode at $\mathcal{K} = 0$ reported in [68]. These numerical procedure is then used to produce the numerical data shown in Figure 4 which shows agreement between hydrodynamic prediction of the speed of sound and the holographic result.

We end this section by explaining what the above numerical procedure means in terms of the correlation function. Consider the near boundary expansion of $Z$, one finds that

$$Z(u) = z_0 + z_2 u^2 + \mathcal{O}\left(u^3\right) \,, \tag{92}$$

with $z_0$ corresponds to the source $\omega \delta a_z + q_z \delta a_t$ in the dual field theory. The coefficient $z_2$ encodes the information of the $\delta j^\mu_{cov}$. Thus the ratio $z_2/z_1$ is related to the following object

$$\frac{z_2}{z_1} \sim \frac{\delta j^\mu_{cov}}{\delta a_\nu} \,, \tag{93}$$

which can be shown, via (7), that it differs from the current-current correlation function $G^{\mu,\nu}_{jj}(\omega, q_z)$ by a contact term. The solution for $\omega, q_z$ which yield non-vanishing $z_2$ with $z_1 = 0$ is then corresponds to the pole of $G^{\mu,\nu}_{jj}$.

## 5 Discussions and Outlook

In this work, we provide a procedure to construct and the resulting hydrodynamic description of a gapless IR theory with 2-group global symmetry. The present constitutive relation has been derived only at the ideal hydrodynamic level, where all dissipative corrections are neglected. For this description to be more reliable, one ought to classify the possible dissipative correction which usually occur at the first order in the derivative expansion. A clear cut scheme to include such effect, as well as statistical fluctuations, from the Schwinger-Keldysh effective action can be found in e.g. [41, 44] (and in [65, 66] from holographic side) and can be readily applied to our setup. Interestingly, despite similarities between the ideal 2-group hydrodynamics and anomalous ideal fluid in $1 + 1$ dimensions, the effect of the dissipation and fluctuations are very different. In the latter case, the statistical fluctuations severely invalidate the usual hydrodynamic expansion scheme (both with and without translational symmetry [35, 70]). On the other hand, these effects are expected to be more controllable for 2-group hydrodynamics as its symmetry structure exists in higher dimensions [35].

We have so far, restricted ourselves in the *normal phase*, where (modulo subtleties in the 1-form case) the symmetries are mostly unbroken. This is clearly not the only possible phase. The 2-group is, after all, a genuine global symmetry and can be explicitly or spontaneously broken as well as anomalous, see [19, 21] for various examples. Study of possible phases and their hydrodynamic descriptions for the higher-form $U(1)$ symmetry has been done in [6], see also [37]. It would be very interesting to do the same for 2-group global symmetry.

We only consider the case where the 2-group composed out of $U(1)$ zero-form and $U(1)$ one-form symmetry but this is by no means the only possible 2-group. Our method is also applicable to the case where the zero-form $U(1)$ is replaced by another Lie group $G$ and the one-form symmetry by a general Abelian group $\mathfrak{a}$. In this case the 2-group structure constant $\hat{\kappa}$ remains to be an integer characterised by the third group cohomology class of $G$ with coefficient in $\mathfrak{a}$, $H^3(BG, \mathfrak{a})$ [21]. A number of interesting gapless strongly interating theories in the IR, with interesting choices of $G$ and $\mathfrak{a}$, can be found in e.g. [16, 19, 21] with one particularly interesting example being the case where the zero-form $G$ is the Poincaré group itself! It would be very interesting to use the hydrodynamic approach to understand their transport and out-of-equilibrium properties.

On an even more abstract level, 2-group is essentially one of many generalised symmetry structure beyond a group, see e.g. [16] for a point of view where multiplications of its elements is not strictly associative or [11] from the category theory perspective. The 2-group structure considered in this work is but one of many of these generalisation (see footnote 4 in the introduction). There are also a generalisation to a higher-group which involves a *two*-form symmetry, appearing even in as seemingly simple a system as axion electrodynamics [71]. It would be very interesting to understand how such larger symmetry structures fit into a hydrodynamic framework. Our work here is a small step toward exploring this landscape of higher-form structures.

Let us also point of some of the more practical future direction. The 2-group structure study here can be obtained by gauging the non-anomalous subgroup of a theory with a very particular mixed anomaly coefficient. In general, a theory with global $U(1)_a \times U(1)_v$ symmetry in $(3+1)$-d, there can be four (consistent) anomaly structure encoded in the anomalous Ward identities in the following way

$$
\begin{aligned}
d \star \langle j_a \rangle &= -\kappa_{a^3} da \wedge da - \kappa_{a^2 v} da \wedge dv - \kappa_{a v^2} dv \wedge dv \,, \\
d \star \langle j_v \rangle &= -\kappa_{v^3} dv \wedge dv \,.
\end{aligned}
\tag{94}
$$

Whenever $\kappa_{v^3} = 0$, one can proceed to gauge the $U(1)_v$ and ask about the global symmetry of the system after gauging. This turns out to be a rather non-trivial and very interesting question; see [18, 72] for some recent work in this direction. We have discussed the case where only $\kappa_{a^2 v} \neq 0$. However an extremely interesting case is when $\kappa_{a v^2} \neq 0$ is also nonzero,[15] as in the case of the usual massless Dirac fermion. In this case, formally the 0-form current is simply not conserved, and the rules of hydrodynamics need not apply. Nevertheless, there exists an effective theory which describe this type of theory called chiral magnetohydrodynamics [24–27] with possible applications in the evolution of the early universe, see e.g. [73]. Though challenging, it would be very interesting to understand (if one exists) the modified global symmetry structure when such other anomaly coefficients are present and systematically derive the hydrodynamic constitutive relations with the principle outlined in this work.

Stepping away from hydrodynamics entirely, we consider future holographic directions. We considered only the Maxwell-type theory where zero-form and one-form symmetry are both $U(1)$. Nevertheless, it should be possible to construct an action where the one-form symmetry is $\mathbb{Z}_n$, similar to those in [60]. In fact, many interesting theories with 2-group global symmetry

---

[15]We thank D. Hofman and U. Gursoy for discussions on this point.

that are strongly coupled in the IR are known to have $\mathbb{Z}_n$ one-form symmetry, such as those in $d+1 = 3$ in [21] and $d+1 = 6$ in [74].[16] At finite temperature and densities, the holographic dual is conceivably the only way to access macroscopic features of these theory and it would be very interesting to explore such possibilities.

# Acknowledgements

We would like to thank J. Armas, A. Brandenburg, L. Delacretaz, D. Dorigoni, D. Friedan, P. Glorioso, U. Gursoy, S. Grozdanov, D. Hofman, A. Jain, J. McGreevy, N. Mekareeya, N. Obers, W. Sybesma for helpful discussions and comments. We are particularly grateful to S. Grozdanov, D. Hofman for comments on the manuscript and especially L. Delacretaz and P. Glorioso for pointing out flawed arguments and incorrect conclusions in an earlier version of the draft.

**Funding information**   The work of N. P. was supported by Icelandic Research Fund grant 163422-052. N. P. would like to thank NORDITA, Durham University, University of Genoa, Niels Bohr Institute and Max Planck Institute for physics of complex systems for their hospitality. N. P. would also like to acknowledge the support from COST Action MP1405 (QSPACE) for supporting his visit to Durham University. N. I. is supported in part by the STFC under consolidated grant ST/L000407/1. All authors thank NORDITA for hospitality during the program "Bounding Transport and Chaos" where our collaboration was initiated.

# A   Entropy production analysis

Here, we show that the additional structure is necessary for 2-group Ward identity in order to have vanishing entropy production. The computation is almost identical to that of the anomalous theory in $1+1$ dimensions.[17]

It is convenient to work on the covariant current $j^{\mu}_{cov}$. We wish to construct the most general constitutive relations in terms of the same set of fluid variables as in fluid with zero-form and one-form $U(1)$ as in [4,43]. Suppose we did not know about the effective action construction, one can assume that the constitutive relation at zeroth derivative level takes the following form

$$
\begin{aligned}
T^{\mu\nu} &= (\varepsilon + p)u^{\mu}u^{\nu} + pg^{\mu\nu} - \mu_b\rho_b h^{\mu}h^{\nu} + \theta\left(u^{\mu}h^{\nu} + u^{\nu}h^{\mu}\right), \\
J^{\mu\nu} &= \rho_b\left(u^{\mu}h^{\nu} - u^{\nu}h^{\mu}\right), \\
j^{\mu}_{cov} &= \rho_a u^{\mu} + mh^{\mu}.
\end{aligned}
\tag{A.1}
$$

For this set of equation to be thermodynamically consistent, we demand that there is an entropy current $s^{\mu}$ which satisfy $s = su^{\mu}$ where $s$ is the thermodynamic entropy and that the entropy production must vanish onshell i.e. $\nabla_{\mu}s^{\mu} = 0$. The entropy current is assumed to be the following

$$
s^{\mu} = \frac{p}{T}u^{\mu} - T^{\mu\nu}\left(\frac{u_{\nu}}{T}\right) - j^{\mu}_{cov}\left(\frac{\mu_a}{T}\right) - J^{\mu\nu}\left(\frac{\mu_b h_{\nu}}{T}\right) + \tilde{s}^{\mu}.
\tag{A.2}
$$

Demanding that $s^{\mu} = su^{\mu}$, we can fix the form of $\tilde{s}^{\mu}$ to be

$$
\tilde{s}^{\mu} = \left(\frac{\mu_a}{T}\right)mh^{\mu} + 2\theta u^{(\mu}h^{\nu)}\frac{u_{\nu}}{T},
\tag{A.3}
$$

---

[16]More recent discussions where 2-group structure made an appearance in the context of 6d QFTs can be found in e.g. [75,76].

[17]We are very grateful to L. Delacretaz for discussions on the content of this section.

where $sT = \varepsilon + p - \mu_a \rho_a - \mu_b \rho_b$. We will not assume a specific form of $\tilde{j}^\mu$ and $\theta$ except that it has to be composed of thermodynamic quantities and $u^\mu, h^\mu$. The entropy production can be written as

$$
\begin{aligned}
\partial_\mu s^\mu = & \left[ \partial_\mu \left( \frac{p u^\mu}{T} \right) - \left( T^{\mu\nu} - 2\theta u^{(\mu} h^{\nu)} \right) \partial_\mu \left( \frac{u_\nu}{T} \right) - \rho_a u^\mu \partial_\mu \left( \frac{\mu_a}{T} \right) - J^{\mu\nu} \partial_\mu \left( \frac{\mu_b h_\nu}{T} \right) \right] \\
& - \left( \partial_\mu T^{\mu\nu} \right) \frac{u_\nu}{T} - \left[ (\partial_\mu j^\mu_{cov}) - \partial_\mu(m h^\mu) \right] \frac{\mu_a}{T} + 2\partial_\mu \left( \theta u^{(\mu} h^{\nu)} \right) \frac{u_\nu}{T} .
\end{aligned}
\tag{A.4}
$$

Here, we will take the metric to be flat but allow nontrivial flux for $a_\mu$ and $b_{\mu\nu}$. Terms in the first line of (A.4) vanish once we impose the thermodynamic relations. We can then focus on the contribution from the second line which, upon imposing the 2-group Ward identity, yield

$$
\begin{aligned}
\partial_\mu s^\mu = & -\left[ \left( H^\nu_{\ \rho\sigma} J^{\rho\sigma} + (da)^\nu_{\ \rho} j^\rho_{cov} \right) \frac{u_\nu}{T} \right] - \left[ \frac{\hat{\kappa} \mu_a}{T} J^{\mu\nu} (da)_{\mu\nu} - \partial_\mu(m h^\mu) \frac{\mu_a}{T} \right] \\
& + \frac{1}{T} \left[ -(h \cdot \partial)\theta + \theta u_\nu (u \cdot \partial) h^\nu - \theta \partial_\mu h^\mu \right] .
\end{aligned}
\tag{A.5}
$$

Substitute the constitutive relation in (A.1) to (A.5), we find that $H^\nu_{\ \rho\sigma} J^{\rho\sigma}$ vanishes and the entropy production reduced into

$$
\begin{aligned}
T \partial_\mu s^\mu = & -(m + 2\hat{\kappa} \mu_a \rho_b) u^\mu h^\nu (da)_{\mu\nu} + (\mu_a m - 2\theta) \partial_\mu h^\mu \\
& + \left( \mu_a \frac{\partial m}{\partial T} - \frac{\partial \theta}{\partial T} - \frac{\theta}{\rho_b} \frac{\partial \rho_b}{\partial T} \right) (h \cdot \partial) T + \left( \mu_a \frac{\partial m}{\partial \mu_a} - \frac{\partial \theta}{\partial \mu_a} - \frac{\theta}{\rho_b} \frac{\partial \rho_b}{\partial \mu_b} \right) \\
& + \left( \mu_a \frac{\partial \mu}{\partial \mu_b} - \frac{\partial \theta}{\partial \mu_b} - \frac{\theta}{\rho_b} \frac{\partial \rho_b}{\partial \mu_b} \right) (h \cdot \partial) \mu_b .
\end{aligned}
\tag{A.6}
$$

Each terms in the above expressions can be shown to be linearly independent from one another. Therefore, the vanishing of entropy production requires all coefficients to vanish. The choice of $m$ and $\theta$ that satisfy such conditions are To cancel the change in entropy

$$
m = -2\hat{\kappa} \mu_a \rho_b , \qquad \theta = \frac{1}{2} \mu_a m = -\hat{\kappa} \mu_a^2 \rho_b .
\tag{A.7}
$$

This agrees with the constitutive relation obtained from the effective action in Eq. (4a)-(4c).

# B  Holographic deconstruction of a theory with one-form $U(1)$ symmetry

Here we discuss the holographic deconstruction of a liquid with 1-form global symmetry. We discuss the effective hydrodynamic description of a holographic theory with bulk action:

$$
S = \int d^{d+2} X \sqrt{-G} (d\mathcal{B})_{MNP} (d\mathcal{B})^{MNP} .
\tag{B.1}
$$

As pointed out in [43], the effective theory is written in terms of a variable $B = b + d\varphi$ with a one-form Stueckelberg field $\varphi = \varphi_\mu dx^\mu$. The background and Stueckelberg $(b, \varphi)$ transformed simultaneously as

$$
b \to b + d\Lambda , \qquad \varphi \to \varphi - \Lambda ,
\tag{B.2}
$$

and the effective action of the field theory dual can only depends on $B = b + d\varphi$. It also enjoy the following one-form chemical shift

$$
\varphi_0 \to \varphi_0 , \qquad \varphi_i \to \varphi_i + C_i(x^j) .
\tag{B.3}
$$

This means that the two Wilson lines along two spacelike curves $L_1, L_2$ denoted by $W(L_1) = \exp\left(i \int_{L_1} dx^i \varphi_i\right)$ and $W(L_2) = \exp\left(i \int_{L_2} dx^i \varphi_i\right)$ are not correlated. This is the same assumption as in the main text.

Let's see how to derive this chemical shift (B.3) from holography.

1. First, we denote that $\mathcal{B}_{\mu\nu}(\infty) \sim b_{\mu\nu}$. Then pick the radial gauge by doing the following transformation

$$
\begin{aligned}
\mathcal{B}_{r\mu} \to \mathcal{B}_{r\mu}^{(1)} &= \mathcal{B}_{r\mu} + \partial_r \varphi_\mu - \partial_\mu \varphi_r , \\
\mathcal{B}_{\mu\nu} \to \mathcal{B}_{\mu\nu}^{(1)} &= \mathcal{B}_{\mu\nu} + \partial_\mu \varphi_\nu - \partial_\mu \varphi_r ,
\end{aligned}
\tag{B.4}
$$

with

$$
\varphi_\mu = -\int dr\, \mathcal{B}_{r\mu} , \qquad \varphi_r = 0 ,
\tag{B.5}
$$

so that $\mathcal{B}_{r\mu}^{(1)} = 0$ in the entire bulk. The boundary value of radial gauge $\mathcal{B}_{\mu\nu}^{(1)}$ is then the combination $B_{\mu\nu} = b_{\mu\nu} + \partial_\mu \varphi_\nu - \partial_\nu \varphi_\mu$.

2. Then we can ask what is the residual gauge transformation that preserve the gauge choice $\mathcal{B}_{r\mu}^{(1)} = 0$. This is nothing but $\varphi_\mu \to \varphi_\mu + C_\mu(x^\mu)$ where $C_\mu$ is radial independent. However, we also have impose the horizon regularity

$$
\mathcal{B}_{0\mu}(r_h) = 0 .
\tag{B.6}
$$

This implies that $\varphi_0 \to \varphi_0$ and $\varphi_i \to \varphi_i + C_i(x^j)$ are the only allowed residual gauge transformation. This gives promised one-form chemical shift in (B.3).

It should be emphasised here that the residual gauge transformation of $\mathcal{B}_{\mu\nu}^{(1)}$ is exactly like the one-form shift symmetry in (53). To see this, let's look at transformation of $b_{0i}$ and $\varphi_\mu$ under this transformation

$$
b_{0i} \to b_{0i} + \underbrace{\partial_0 C_i}_{=0} - \partial_i C_0 , \qquad \varphi_\mu \to (\varphi_0, \varphi_i + C_i) .
\tag{B.7}
$$

Let us also comment on the construction of [6] which is more closely related to the construction of equilibrium partition function in Section 2.1. There, they interpret the fact that $\varphi_0$ which does not transformed under the chemical shift as an indication that it is a Goldstone. Their fomulation can be written in a simpler form when one consider the equilibrium configuration with the spacetime being $\mathbb{R}^d \times S^1$. One can define a one-form chemical potential as a vector $\mu_i$ by intergrating $b_{\tau i}$ along the thermal cycle (analogous to those in [48, 49], it is not invariant but transformed as

$$
\frac{\mu_i}{T} = \int_{\tau=0}^{1/T} d\tau\, b_{\tau i} = \int_{\tau=0}^{1/T} d\tau\, u^\mu b_{\mu i} \quad \to \quad \frac{\mu_i}{T} - \partial_i \int d\tau \Lambda_\tau ,
\tag{B.8}
$$

unlike the zero-form case where $\mu/T = \int d\tau a_\tau$ which is perfectly invariant (One may also assume that all the fields are independent of $\tau$ and use the definition $\mu_i = u^\mu b_{\mu i}$ with $u^\mu = (1, \mathbf{0})$. But it essentially boils down to the same thing). To fix this they introduce a Goldstone $\psi$ which transformed as

$$
\psi \to \psi - \Lambda_\tau(x^i) , \qquad \text{while} \qquad b \to b + d\Lambda .
\tag{B.9}
$$

Then, the combination

$$
\int d\tau \, (b_{\tau i} - \partial_i \psi) = \int d\tau \left(u^\mu b_{\mu i} - \partial_i \psi\right)
\tag{B.10}
$$

is invariant and they use this to define the one-form chemical potential. One can try to connect it to the field $\varphi_\mu$, we ask ourselves what is the transformation property of $\varphi_\mu$ such that

$$-\partial_i \psi = u^\mu \left( \partial_\mu \varphi_i - \partial_i \varphi_\mu \right) = \partial_0 \varphi_i - \partial_i \varphi_0 \,. \tag{B.11}$$

So that the chemical potential is defined to be $\mu_i = b_{0i} + \partial_0 \varphi_i - \partial_i \varphi_0$ as in [43]. Note that this definition of chemical potential is redundant as the redefinition of $\varphi_\mu$

$$\varphi_0 \to \varphi_0 \,, \qquad \varphi_i \to \partial_i C(x^i) \,, \qquad b_{\mu\nu} \quad \text{fixed} \,, \tag{B.12}$$

does not change the chemical potential. The above transformation, again, is nothing but the one-form shift in (53).

## C Useful formulae

These identities are useful to obtain the constitutive relation (4a)-(4c) from the effective action obtained at the end of Section (3.1)

$$\delta T = \frac{T}{2} u^\alpha u^\beta \delta g_{\alpha\beta} \,, \tag{C.1}$$

$$\delta u^\mu = \frac{1}{2} u^\mu u^\alpha u^\beta \delta g_{\alpha\beta} \,, \tag{C.2}$$

$$\delta \Delta^{\mu\nu} = \left( -g^{\mu\alpha} g^{\nu\beta} + u^\mu u^\nu u^\alpha u^\beta \right) \delta g_{\alpha\beta} \,, \tag{C.3}$$

$$\delta \mu_a = \frac{\mu_a}{2} u^\alpha u^\beta \delta g_{\alpha\beta} + u^\alpha \delta a_\alpha \,, \tag{C.4}$$

$$\delta A_\mu^\perp = \frac{\mu_a}{2} \left( \Delta^\alpha_{\ \mu} u^\beta + \Delta^\beta_{\ \mu} u^\alpha \right) \delta g_{\alpha\beta} + \Delta^\alpha_{\ \mu} \delta a_\alpha \,, \tag{C.5}$$

where $\Delta^{\mu\nu} := g^{\mu\nu} + u^\mu u^\nu$ and $A_\mu^\perp := \Delta_\mu^{\ \nu} A_\nu$ is the invariant combination $A_\mu$ projected onto a plane orthogonal to $u^\mu$. Then, the variation of the chemical shift invariant $\mu_b h_\nu = u^\nu B_{\nu\mu} - \hat{\kappa} \mu_a A_\mu^\perp$, where $h_\mu h^\mu = 1$, can be written as

$$\delta(\mu_b h_\mu) = \frac{1}{2} \left( \nu_\mu u^\alpha u^\beta - \hat{\kappa} \mu_a^2 \left( u^\alpha \Delta^\beta_{\ \mu} + u^\beta \Delta^\alpha_{\ \mu} \right) \right) \delta g_{\alpha\beta} - u^\nu \delta b_{\mu\nu} - \hat{\kappa} \mu_a \Delta_\mu^{\ \nu} \delta a_\nu \tag{C.6}$$

$$- \hat{\kappa} A_\mu^\perp u^\nu \delta a_\nu + \hat{\kappa} u^\nu \phi \left( \partial_\nu \delta a_\mu - \partial_\mu \delta a_\nu \right) \,,$$

$$\delta \mu_b = h^\alpha \delta(\mu_b h_\alpha) - \frac{\mu_b}{2} h^\alpha h^\beta \delta g_{\alpha\beta} \,. \tag{C.7}$$

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
