# Peer review of "-group global symmetries, hydrodynamics and holography"

_SciPost Physics, doi:SciPost Phys. 15, 063 (2023)_

## Round 1 · Referee Report · Anonymous · 2022-4-19

Report

This manuscript is concerned with the hydrodynamic regime of QFTs with 2-group symmetries---a particular current algebra involving 0-form and 1-form currents.The recent accounting of higher-form symmetries in hydrodynamics has led to a more systematic description of the finite temperature dynamics of systems including magnetohydrodynamics [Ref. 4]. Anomalies for continuous global symmetries moreover are known to non-perturbatively fix certain transport parameters [e.g. Ref. [30]]. Given the difficulty of otherwise constraining transport in interacting systems, it is important to further establish other non-perturbative consequences of nontrivial current algebras, which the authors undertake.

The results of this paper -- some of which are verified in holographic constructions -- seem correct. One consequence of the 2-group structure that the authors find is the chiral propagation of the collective mode associated with the 0-form symmetry, along flux lines associated with the 1-form symmetry, which themselves are dynamical.

Requested changes

None

---

## Round 1 · Referee Report · Anonymous · 2022-5-14

Report

The paper develops the formalism to study the state of thermal equilibrium in a fluid that is endowed with a 2-group global symmetry structure. The authors study a fluid with an ordinary 1-form current intertwined with a conserved 2-form current via the 2-group Ward identities. They construct the most general equilibrium constitutive relations for such a fluid. In particular, they provide an impressive detailed exposition of the construction of the equilibrium partition function, which is invariant under diffeomorphisms as well as the gauge transformations of the 1-form and 2-form background gauge fields which couple to their respective currents, by introducing Stueckelberg fields into their setup. The equilibrium partition function, which gets further constrained due to the shift symmetries associated with the Stueckelberg fields, gives rise to the aforementioned constitutive relations when varied with respect to the background sources. Additionally, the authors also discuss deriving the same equilibrium constitutive relations via an entropy current analysis in Appendix A of the paper, lending further credibility to their equilibrium partition function approach.

Furthermore, the authors discuss some of the interesting hydrodynamic modes of the system, along with their physical interpretation. Section IV then proposes a working holographic dual to their setup, which is able to capture these hydrodynamic modes faithfully. In particular, in their holographic setup, the authors do a very nice job of clearly explaining how the shift symmetries of the Stueckelberg fields of the boundary theory are realized in the radial gauge in the dual bulk description.

The paper opens up the possibility of exploring several new avenues, including but not limited to the study of 2-group symmetries in hydrodynamics with other constituent group structures; the possibility of breaking the 2-group symmetry structure spontaneously or explicitly and studying the ensuing affects on hydrodynamic transport; going beyond the ideal order and exploring dissipative corrections to the 2-group Ward identities and constitutive relations etc. All in all, the paper is an interesting addition to the existing literature on hydrodynamics with higher-form symmetries, which have received much attention lately.

There are a couple of questions which I would like the authors to comment upon in the paper, which would be helpful for a better understanding to the reader.

1. One of the key results of the paper is the existence of a 1-form current along the direction of the magnetic field, even in an equilibrium state where the fluid is chosen to be at rest i.e. $u^\mu = (1, \vec{0})$. This is somewhat surprising, because if the fluid is at rest, then what is it that is flowing along the direction of $h^\mu$ in thermal equilibrium? In chiral magnetic effect, there too is a current flow along the direction of the magnetic field, but it is an $\mathcal{O}(\partial)$ effect, and not an equilibrium effect. The presence of a current of charged particles along the magnetic field when the fluid is at rest appears to be a mutually contradictory conclusion. It would be nice if the authors can provide a clean physical picture of what is happening here.

2. The concrete holographic model which the authors utilize for their calculations in Section IV-B involves a 1-form and a 2-form gauge field propagating on an AdS$_5$-Schwarzschild background spacetime. Thus they neglect the back-reaction entirely. Why is this approximation justified? If one is working with a strongly coupled ordinary $U(1)$ charged fluid, without any higher-form symmetry, the dual holographic description is provided by electrically charged black brane solutions to Einstein-Maxwell theory (or Einstein-Maxwell-Chern-Simons theory if the boundary theory has a chiral anomaly) in thermal equilibrium. For the setup relevant to the current paper, one would expect a dual bulk spacetime geometry with both a 0-form and 1-form charge density on the black brane. Are such geometries discussed in the literature? It would be nice if the authors comment about the necessary limitations of their holographic analysis and the ensuing results, by not looking at the correct dual background geometry, but rather just a probe approximation to it.

Finally, some typos: The indices in eqs. (1.8a) and (1.8b) should be corrected, and factors of $\hat{\kappa}$ missing in eqs. (4.8), (4.9) and (4.10) should be restored.

  • validity: -
  • significance: -
  • originality: -
  • clarity: -
  • formatting: excellent
  • grammar: below threshold

---

## Round 2 · Referee Report · Anonymous (Referee 1) · 2023-5-18

Report

I believe the manuscript is still publishable, despite the improvements made after the first round of reviews.

---

## Round 2 · Referee Report · Anonymous (Referee 2) · 2023-5-23

Report

With the additional clarifications and discussion added by the authors, I am now happy to recommend the manuscript for publication.

---

## Round 2 · List of Changes

We would like to thank the referee for their comments and encouraging message, and we apologize for the long delay in our response. Regarding the comments requested by Referee 2, we have adjusted the manuscript as follows

  1. We agree with the referee that the existence of a charged current even in equilibrium is an interesting phenomenon: we have added some further clarifying words about this on page 7, explaining further their microscopic origin and giving some other examples from elementary physics. At some level the microscopic origin is precisely the same as in the regular chiral magnetic effect – the key difference is just that the magnetic field itself is now a 0th order hydrodynamics degree of freedom, as we discuss further in the paper.

  2. Below Eq. (4.14), we add a quick explanation of the limit in the dual QFT in which the backreaction is negligible and comments on the limitation of the presented background solution. We are not aware of the fully backreacted black brane solution but it is likely that it has to be obtained numerically (similar to those with finite 1-form charge density).

  3. We have corrected the typos pointed out by the referee as well as a few other grammatical and spelling mistakes.

We hope that with these additions and corrections, the manuscript now meets the standard of SciPost.

---

## Editorial Decision

published